# The Roles of Mating, Age, and Diet in Starvation Resistance in *Bactrocera oleae* (Olive Fruit Fly)

**DOI:** 10.3390/insects14110841

**Published:** 2023-10-29

**Authors:** Evangelia I. Balampekou, Dimitrios S. Koveos, Apostolos Kapranas, Georgios C. Menexes, Nikos A. Kouloussis

**Affiliations:** School of Agriculture, Aristotle University of Thessaloniki, 54124 Thessaloniki, Greece; evibal@agro.auth.gr (E.I.B.); koveos@agro.auth.gr (D.S.K.); akapranas@agro.auth.gr (A.K.); gmenexes@agro.auth.gr (G.C.M.)

**Keywords:** stress, aging, food type, lifespan, pest management, sustainability, *Bactrocera oleae*, olive fruit fly

## Abstract

**Simple Summary:**

The olive fruit fly (*Bactrocera oleae* (Rossi) (Diptera: Tephritidae)) is a pest of major economic importance that threatens the olive industry. Studying several factors affecting the survival ability of this insect during food deprivation, such as its mating status, age, and diet, may provide important insights into the biology of *B. oleae* that are useful for its effective control. The starvation resistance (hours of survival after the removal of food) of adult olive fruit flies was measured in four age classes in virgin and mated adults fed a full diet (water/sugar/yeast hydrolysate as protein in a 5:4:1 ratio) or a restricted, sugar-based diet lacking in protein, examining both males and females. The pattern of starvation resistance was the same for both genders under the same conditions (mating status, age, and diet) in the laboratory. Specifically, (a) mated adults showed much less resistance to starvation compared to virgin adults; (b) younger adults endured longer starvation periods compared to older adults; and (c) adults fed the restricted diet endured longer starvation periods than those fed the full diet. We conclude that mating, a full diet, and aging reduce starvation resistance.

**Abstract:**

The olive fruit fly (*Bactrocera oleae* (Rossi) (Diptera: Tephritidae)), although a pest of major economic importance for the olive industry, has not been sufficiently studied with respect to the factors affecting its survival resistance to food deprivation. In the present study, we examined the effect of the interaction between mating status (virgin/mated), age class (11–20/21–30/31–40/41–50), and diet quality (protein plus sugar or only sugar) on starvation resistance in *B. oleae* under constant laboratory conditions. We conducted a total of 16 treatments (2 × 4 × 2 = 16) for each gender. Our results showed that starvation resistance in *B. oleae* did not differ significantly between females and males. The main conclusions of our study regarding mating status, age, and diet indicated that mated adults showed much less starvation resistance compared to virgins, younger adults endured longer, and the adults fed a restricted diet endured longer than those fed a full diet. A three-way interaction between mating status, diet, and age class was also identified and was the same for both genders. The interaction between mating status, age class, and diet also had a significant influence on starvation resistance in both sexes.

## 1. Introduction

*Bactrocera oleae* (Rossi) (Diptera: Tephritidae) (olive fruit fly) is a pest of major importance in the olive fruit production industry, as it causes up to 30% of the yield damage to olive crops worldwide (along with fungi and weeds) [1,2]. Female *B. oleae* lay their eggs inside the olive fruit [3], reducing the quality of both the olive fruit and the olive oil produced [4]. Controlling the olive fruit fly is difficult because the larvae feed inside the olive fruit, in a protected environment; therefore, pest control can be effective only before oviposition takes place. Given the potential losses of olive crops due to *B. oleae*, olive farmers in the past constantly used conventional chemical insecticides to reduce yield losses [5]. For many years, the use of insecticides was the only available approach to suppressing the population of insects [1,2]. However, several concerns were soon raised concerning an increase in the insects’ resilience to chemical substances, as well as the health of humans or other mammals due to the presence of pesticide residues, which were very often detected in olive oil [6]. In addition, it has been reported that the use of insecticides has a negative effect on the natural enemies of *B. oleae*, such as *Chrisopids* (Neuroptera: Chrysopidae) [7] and *Psyttalia concolor* (Szépligeti) (Hymenoptera: Braconidae) [8]. To address these issues related to the use of chemicals, most olive-growing countries have adopted the concept of integrated pest management (IPM) as a sustainable strategy for olive crop protection to reduce the amounts of chemicals used to control pests in olive groves, and they constantly seek other methods consistent with economic, ecological, and toxicological requirements to maintain pests below the economic threshold while giving priority to natural limiting factors [9].

Lately, there has been an increased interest in exploring more sustainable control methods, such as the sterile insect technique (SIT) [10,11]. The efficacy of this method relies on producing mass-reared male insects, which, when released into the wild in large numbers, are more competitive than wild males and are also capable of greater endurance under stressful environmental conditions [10,11], such as food deprivation. Starvation resistance (SR) is considered an important trait in pest management [12,13,14,15,16]; starvation resistance and thermal stress [12] have been identified as two of the most common environmental conditions that insects may face in their lifetimes [17]. Key factors that affect the starvation resistance in insects are their mating status, age, and diet [15,16,18,19,20]. For males, mating or even only courtship can lead to significant energy expenditure and consequently shorten the insect’s lifespan [21,22,23,24]. Females mainly face energy losses due to egg maturation [18]. An insect’s mating status is also influenced by environmental conditions and strain [25,26,27]. So far, the study of aging in insects has shown that, regarding their starvation resistance, their survival ability decreases with age [14,16,28]. Starvation resistance in adult insects fed different diets affects their fitness [29]. Survival and sexual signaling were shown to be crucially influenced by diet quality early in a male’s life [24]. The evaluation of insect survival under varying conditions of food availability, diet, and quality provides insights into the factors that drive the evolution of different feeding strategies and helps us better understand the biology and ecology of insects [14,29]. The effects of food deprivation have been studied very little in model organisms, such as *Drosophila melanogaster* (Meigen) (Diptera: Drosophilidae) [30,31], and insects of agricultural importance, like *Ceratitis capitata* (Wiedemann) (Diptera: Tephritidae) [16]. A recent study on *D. melanogaster* showed that feeding adults a protein-based diet for twenty generations led to lower body weights and wing reductions in male adults [32], and the trade-off between reproduction and lifespan is still an area under investigation [33]. There are several studies on topics related to *B. oleae* regarding the physiology of the insect (e.g., the determination of volatile substances in olives and their effect on reproduction [34], mating competition between wild and artificially reared adults [35] and altered activity and rest patterns [36]), but, to our knowledge, starvation resistance in *B. oleae* has not yet been studied. Additionally, the only existing work in the literature addressing starvation resistance in Tephritidae that examines how aging and diet affect starvation resistance was conducted on virgin *C. capitata* [16], which is a polyphagous, cosmopolitan insect. There are some studies on the topic of starvation resistance in Tephritidae, evaluating the effect of diet on several physiological aspects of the adult olive fruit fly, such as lipid reserves, the onset of oviposition, lifetime egg production and the longevity of adults [37], field survival [38], male sexual performance in relation to insect vulnerability to starvation [38,39], or starvation resistance in different time intervals [40], but none of them focus on the interaction between different parameters and how they influence adults’ resistance to starvation. Furthermore, useful insights on starvation resistance in insects and the importance of factors affecting their adaptation to periods of food scarcity have already been identified in studies on *D. melanogaster* [41].

The olive fruit fly is an oligophagous insect species relying on olives for its survival and reproduction [3]. Thus, the survival of this species depends solely on the olive fruit. Additionally, the olive tree commonly produces a much greater-than-average crop in one year and a much lower-than-average crop in the following year in olive cultivation [42]. Therefore, these features, in conjunction with starvation resistance in *B. oleae*, as a result, shape the population dynamics of the olive fly [43]. During years characterized by low olive yields, it becomes crucial for the olive fruit fly to live for a longer period and, therefore, to maximize its reproductive potential by producing a greater number of offspring [42,43]. In the above context, the effects of mating status, age, and diet on starvation resistance in *B. oleae* were studied for both males and females. The aims of this study were to identify which mating status (virgin, mated), age class in days (11–20, 21–30, 31–40, 41–50), and diet (full, restricted) make *B. oleae* more vulnerable, namely, more susceptible to several stresses, and to study the effect of the interaction between these three factors on starvation resistance in this pest, namely, the combination of values of the three factors that resulted in increased insect vulnerability due to starvation. We hope that studying the key factors affecting *B. oleae* under food deprivation will provide insights that will help improve existing pest control management or even formulate new, more sustainable and effective strategies.

## 2. Materials and Methods

The experiment was conducted in three main stages (Figure 1): (1) collecting non-infested olives for oviposition and infested olives to obtain adults for the experiments and maintain the colony; (2) preparing experimental insects that would be used in the experiments; (3) recording the deaths every 4 h daily from the 11th day up to the 50th day after the experimental insects were subjected to starvation (80 insects daily, with 3200 insects in total).

### 2.1. Origin and Handling of Experimental Insects and Olive Fruit

Adult survival following food deprivation, which is an index of starvation resistance, was assessed in stable laboratory conditions (temperature: 25 ± 2 °C; relative humidity: 65 ± 5%; and photoperiod: L14:D10). Larvae were reared in (non-infested) olive fruit collected in the wild from olive trees free of pests and diseases. The original total number of insects involved was approximately 20,000 adults, while the final number of insects participating in the study (due to the insects’ mortality) was 3200 flies. In all cases, adults were carefully transferred using an aspirator, with particular attention paid to not disturbing the insects. The aspirator was used as a transportation means, allowing insects to enter and exit (from the same point), almost walking, without any force being applied to them. In the case that any insect suffered any disturbance or injury during transportation, it was removed and replaced.

The stages for acquiring and handling the olive fruit were the following:(a)*Insect cage types used in the experiments:* The wild adult olive fly lines were housed in custom-made insect cages based on the model of BugDorm cages (Model DP1000B) usually used in entomological experiments [44]. The types of insect cages used in the laboratory for the needs of the experiments were (A) BugDorm-type cages, 30 × 30 × 30 cm^3^ (colony cages for rearing of the olive fruit fly) (Figure 2a); (B) transparent plexiglass cages with dimensions 20 × 20 × 20 cm^3^ (Figure 2b), where (i) pupae gathered from infested olives in basins were placed into Petri dishes and then were transferred into the plexiglass cages for the adults to emerge (see experimental procedures and protocol for further information), and (ii) flies on the 10th day of their lives were transferred to mate or to be together with other flies of the same sex; (C) individual plastic cages (Figure 2c), where (i) flies were transferred individually upon their emergence, with either full or restricted diet and water, and (ii) flies were transferred at the appropriate age, each time to a new individual plastic cage that was thoroughly clean of any trace of food to measure the hours until death (starvation resistance).(b)*Diet food types for adult fruit flies: Two different diets were used:* (A) a full diet consisting of a mixture of hydrolyzed yeast (protein) in a ratio of 5:4:1 (water/sugar/yeast hydrolysate as protein) or (B) a restricted diet containing only sugar but deprived of protein. Water was supplied to all cohorts through a wetted cotton wick.(c)*Harvesting (non-infested) olive fruit for the rearing of olive fruit flies*: Olive fruits used in the experiments were collected from olive groves located in the region of Chalkidiki and Northern Greece. The olives were selected one-by-one by hand in the above regions from trees that were as free from pests and diseases as possible. To maintain the number of insects needed for the experiment and the genetic diversity of the experimental insect population close to that of natural populations, infested olives were constantly collected from olive groves for a period of approximately three months, and wild insects were constantly introduced to the colony. The total amount of olives needed for the experiment was roughly 200 kg (Figure 3).(d)*Maintaining the (non-infested) olive fruit flies*: Immediately after harvesting, the olive fruits were placed in glass jars in the refrigerator at 6 ± 1 °C (Figure 3).(e)*Collecting infested olive fruits:* Mc Cain traps with an appropriate food attractant were used in the aforementioned regions to identify the period of the first adult flights and the onset of infestation in the field. Olive fruits that had been infested by the olive flies were collected from the trees and transported to the laboratory (Figure 3).

### 2.2. Experimental Design

The response variable measured in the experiments was the duration of time for which the insects survived after food deprivation (starvation resistance, measured in hours). Insect deaths were recorded every 4 h daily by the same human observer. The starvation resistance was then calculated based on the date and time of death. The experimental units that were studied individually for starvation resistance were 3200 adult olive fruit flies (10 adults ×40 days ×2 diets ×2 mating status ×2 gender = 3200) that originated from the larvae in the infested olives. The initial number of flies was at least sixfold that of the flies used in the experiments. This was due to the mortality of the flies observed before they were included in the experiment. All adults that took part in each treatment were derived from larvae reared in olive fruit. The experiments on starvation resistance were carried out from the eleventh to the fiftieth day of the insect’s life. The treatments examined were the following: (a) mating status factor with two levels (virgin and mated); (b) age factor with four levels (age classes: 11–20, 21–30, 31–40, and 41–50 days); and (c) diet factor with two levels (full and restricted diets). There were 16 combinations of treatments in total (2 × 4 × 2 = 16) for each gender. There were also 10 replicates, namely, 10 insects of the same gender and same mating status and at the same age (day of life), also fed the same diet, that were randomly selected for each treatment, as described in detail in the experimental procedures and protocol. The wild *B. oleae* is mature for mating after the 7th day [35]; this is why the 10th day was selected for mating. The period from the 11th to the 50th day was selected, because after an age of 50 days, it is difficult to maintain the number of insects needed for each treatment.

The aim of the experimental design was to measure starvation resistance in olive fruit flies under different conditions (mating status, food/diet, and age) to identify the status at which insects are more vulnerable or more durable. 

### 2.3. Experimental Procedures and Protocol

The experimental processes followed can be grouped into the following stages:(a)*Rearing the parents of the experimental insects:* Adults from infested olive fruits that had been collected from trees hatched inside wooden cages with plenty of water and protein food. After the completion of their hatching, the olives of their origin were removed, and fresh olives were added (ones that we had collected and maintained in the refrigerator). After mating, the females laid their eggs in the olive fruit. These infested olives were removed from the cages, laid into basins, and covered with a suitable cloth to ensure the appropriate humidity and temperature conditions (Figure 3). After pupation and before the emergence of the adults, pupae were transferred to plexiglass cages with dimensions 20 × 20 cm, awaiting the appearance of the adults (Figure 2b). In these plexiglass cages, there was either a full or restricted diet and water (Figure 4).(b)*Handling the experimental insects before the experiment*: Upon emergence, adults were placed in individual plastic cages (Figure 2c) with water and food (either the full or restricted diet). At the age of 10 days, groups of 10 adults of either only females and males (both virgin) or 5 virgin males and 5 virgin females (mated) were allowed to be together in larger cages (20 × 20 cm) for one day (Figure 2b) before being placed back into individual cages (Figure 2c). After this period, the flies were placed back into individual cages (Figure 2c) to eliminate crowding and social interactions (Figure 4). Flies that had been kept with conspecifics of the opposite sex were monitored by a human observer to verify mating. We observed the flies for mating from 16:00 to 21:00 because, in this species, mate searching and courtship take place during the late evening [34,36]. Individuals that had not mated were removed from the experiment and were replaced with others that had mated.(c)*Preparing the flies to undergo starvation (food deprivation)*: The steps followed (Figure 4): (1) initially, experimental insects (pupae) were placed in Petri dishes in plexiglass cages; (2) upon adult emergence, they were transferred individually to plastic cages with water, half with the full diet and half with the restricted diet; (3) on the 10th day, all insects were transferred to 8 plexiglass cages: (a) 4 cages with the full diet (40 adults in total: 1 cage with 10 males, 1 with 10 females, and 2 cages with 5 males and 5 females in each cage) and (b) 4 cages with the restricted diet (40 adults in total: 1 cage with 10 males, 1 with 10 females, and 2 cages with 5 males and 5 females in each cage); (4) at the end of the 10th day, insects were transferred back to their individual cages with water and with the same diet that they were fed in the plexiglass cage.(d)*Recording of deaths—Calculating starvation resistance*: Upon reaching the eleventh day of their adult life, ten individual adults from each treatment at a specific age (11th, 12th, up to 50th day of life) were each transferred to a new individual plastic cage (Figure 2c) thoroughly clean of any trace of food [16]. The insects’ deaths were recorded every four hours due to food deprivation during the light period (four times per day: 08:00, 12:00, 16:00, 20:00). In Figure 5, a schematic representation showing the feeding and starvation stages is given. From the 11th day up to the 50th day of their lifespan, 80 insects (3200 in total) fed the full or restricted diet (40 adults fed the full diet and 40 adults the restricted diet, in each case: 10 virgin males, 10 virgin females, 10 mated males, and 10 mated females) were subjected to starvation in new clear individual cages. Within the period from the 11th to the 50th day, every four hours, the deaths were recorded as a measure of starvation resistance. In case there was difficulty in identifying an insect’s death, a fine paintbrush was used to gently move the insect and confirm its death. Rotation of the plastic cages was performed daily to reduce potential experimental errors. Starvation resistance was finally calculated as the difference between the date and time of death and the date and time of the moment the insects were subjected to food deprivation.

We adopted an age clustering of the results (starvation resistance—time in hours to death) on a ten-day basis, as was presented in similar work on Tephritidae [16], to simplify the results’ interpretation. The experiments conducted were analyzed for each gender individually, because the inherent differences in the effects of mating and fecundity on aging and longevity are usually studied separately for males and females due to their physiology [45]. Additionally, the same pattern of starvation resistance has been identified for both genders.

### 2.4. Statistical Analyses

The survival resistance of adult insects for each gender was analyzed with the ANOVA method within the methodological framework of General Linear Models. The ANOVA model included the effects (three main effects, three two-way interactions, and one three-way interaction) of three factors: the mating factor with two levels (virgin and mated), the diet factor with two levels (protein-rich food and sugar-only food without protein), and the age factor with four levels (age classes: 11–20, 21–30, 31–40, and 41–50 days) [16]. There were 16 combinations in total of the three factors’ levels (2 × 4 × 2 = 16). The ANOVA method was mainly used to estimate the correct standard errors of the differences between the mean values of the factor levels’ combinations. Tukey’s multiple-comparisons procedure [46] was used to test the significance of the differences between the compared mean values. Linear models’ residuals were tested for normality and homoscedasticity. The residuals’ normality assumption was examined by visually inspecting the corresponding histogram and boxplot, comparing the residuals’ median values with the value of 0 (zero), assessing the corresponding skewness and kurtosis indices, and analyzing the results of the Kolmogorov–Smirnov test for normality. The homoscedasticity assumption was examined by visually inspecting the residuals’ scatter plot against the model’s predicted values and assessing the magnitude of Spearman’s rho rank correlation coefficients between the residuals’ absolute values and the model’s predicted values. No serious violations of these two assumptions were detected. Data are presented as mean ± standard error (SE). Additional descriptive statistical indices are presented in Appendix A. Since only the terminal survival time of the insects was recorded, there was no specific need to examine the data using a survival analysis model. All statistical analyses were performed with the IBM SPSS Statistics ver. 26.0 Software (IBM Corp., Armonk, NY, USA). The significance level in all statistical hypothesis testing procedures was preset at *a* = 0.05 (*p* ≤ 0.05).

Finally, we calculated the differences (in percent) between the values (days for which insects endured starvation) for each of the 16 treatments (mating status: virgin, mated) × (age class: 11–20, 21–30, 31–40, 41–50) × (diet: full, restricted) and the grand mean of all flies in each gender regardless of their mating status, age class, and diet to better represent the influence of each of the three factors on insect starvation resistance based on the following formula (Equation (1)):(1)$$\left({\bar{X}}_{T}-\bar{X}\right)*100/\bar{X}$$
where ${\bar{X}}_{T}$ is the mean starvation resistance (in hours) for each treatment, and $\bar{X}$ is the total mean for each gender (in hours).

## 3. Results

### 3.1. Study of Starvation Resistance in Males

Tukey post hoc was applied to examine whether statistically significant differences exist between the treatments (Appendix A). Also, descriptive statistics and the ANOVA results are also available (Appendix A, respectively). There is a significant three-way interaction in males between the mating status, the diet, and the age class, as indicated by ANOVA (*F*(3, 15) = 6.5, *p* < 0.001); see also Appendix A for a summary of the full model ANOVA results. Consequently, the focus is mainly on examining this interaction using the simple–simple effects analysis approach. More specifically, the two-way interaction between the four age classes and the two mated statuses was examined within each diet [47]. The comparison between age classes is based on the mean starvation resistance of the insects measured in hours to death (values in parentheses).

In virgin males fed the full diet (protein), resistance to starvation decreased with age (Figure 6A, Appendix A). In the 11–20 age class, virgin males fed the full diet lived longer compared to the 31–40 age class (in the 11–20 class: 72.0 ± 3.6 h; in the 31–40 class: 48.5 ± 3.2 h; *p* = 0.003) and also compared to the 41–50 age class (in the 41–50 class: 39.6 ± 1.4 h, *p* < 0.001). Insects in the 21–30 age class lived longer than those in the 41–50 age class (in the 21–30 class: 63.3 ± 5.7 h; in the 41–50 class: 39.6 ± 1.4 h; *p* = 0.002). Therefore, in virgin males fed the full diet (protein), the ability to resist stress (starvation resistance) tended to decline as they aged. On the contrary, in mated males fed the full diet, there was no decline in starvation resistance as they aged. In mated males fed the restricted diet (sugar), resistance to starvation decreased with age, following a similar pattern to the one identified for virgin males. Insects in the 11–20 age class lived longer than those in the 31–40 age class (in the 11–20 class: 65.8 ± 4.6 h; in the 31–40 class: 33.8 ± 3.1 h; *p* < 0.001). In virgin males fed the restricted diet (sugar), no decline in resistance was observed as they aged (Figure 6B).

In the following age classes, virgin males fed the full diet were more resistant than mated ones. Specifically, in the 11–20 age class, virgin males fed the full diet were more resistant to starvation than mated ones (in the 11–20 class, virgins: 72.0 ± 3.6 h; mated: 41.0 ± 5.5 h; *p* = 0.001), and in the 21–30 age class, virgin males fed the full diet were more resistant to starvation than mated ones (in 21–30 class, virgins: 63.3 ± 5.7 h; mated: 27.2 ± 1.8 h; *p* < 0.001). In the 31–40 age class, virgin males fed the full diet were more resistant to starvation than mated ones (in the 31–40 class, virgin males: 48.5 ± 3.2 h; mated males: 25.1 ± 2.5 h; *p* = 0.003) (Figure 6A). In the 31–40 age class, virgin males fed the restricted diet were more resistant than mated ones (virgins: 64.0 ± 6.7 h; mated: 33.8 ± 3.1 h; *p* < 0.001). No differences were observed between virgin and mated males fed the restricted diet in the other three age classes (Figure 6B).

In the following age classes, mated males that were fed the full diet were less resistant than those fed the restricted diet. Specifically, in the 11–20 age class, mated males that were fed the full diet were less resistant to starvation than mated adults fed the restricted diet (males on full diet: 41.0 ± 5.5 h; males fed the restricted diet: 65.8 ± 4.6 h; *p* = 0.05), and in the 21–30 age class, mated males fed the full diet were more resistant to starvation than mated ones that were fed the restricted diet (mated males fed the full diet: 27.2 ± 1.8 h; mated males fed the restricted diet: 53.3 ± 4.8 h; *p* < 0.001). On the contrary, no significant differences in starvation resistance were observed between virgin males that received a full diet and those that followed a restricted diet in any age group (Figure 6A,B).

### 3.2. Study of Starvation Resistance in Females

Tukey post hoc was applied to examine whether statistically significant differences exist between the treatments (Appendix A). Also, descriptive statistics and the ANOVA results are also available (Appendix A, respectively). There is a significant three-way interaction in females between mating status, diet, and age class, as determined by ANOVA (*F*(3, 15) = 3.6, *p* = 0.016); see also Appendix A for a summary of the full model ANOVA results. Consequently, the focus is mainly on examining this interaction using the simple–simple main effects analysis approach. More specifically, the two-way interaction between the four age classes and the two mating statuses was examined within each diet [47]. The comparison between age classes is based on the mean starvation resistance of the insects, measured in hours to death (values in parentheses).

In both virgin and mated females fed the full diet, the resistance to starvation decreased with age (Figure 7A, Appendix A). In both cases, females in the 11–20 age class lived longer compared to the 41–50 age class. Specifically, in the 11–20 age class, virgin females fed the full diet lived longer compared to full-diet-fed virgins in the 41–50 age class (in the 11–20 class: 74.4 ± 2.4 h; in the 41–50 class: 46.5 ± 1.5; *p* = 0.001). In the 11–20 age class, mated females fed the full diet (mean starvation resistance in hours to death: 57.4 ± 5.0 h) lived longer compared to their counterparts in the 41–50 age class (mean starvation resistance in hours to death: 32.0 ± 4.6 h, *p* = 0.05). Therefore, in both virgin and mated females that were fed the full diet (protein), their ability to resist stress (starvation resistance) tended to decline with age.

Furthermore, in both virgin and mated females fed the restricted diet, the resistance to starvation decreased with age. Specifically, virgin females in the 11–20 age class fed the restricted diet exhibited significantly higher resistance to stress compared to those in the 31–40 age class (virgin females in the 11–20 class: 107.8 ± 3.7 h; in the 31–40 class: 69.0 ± 4.8 h; *p <* 0.001). In the 11–20 age class, mated females fed the restricted diet exhibited significantly higher resistance to stress compared to those in the 31–40 age class (mated females in the 11–20 class: 78.9 ± 7.8 h; in the 31–40 class: 42.3 ± 2.7 h; *p <* 0.001) and also to those in the 41–50 age class (in the 41–50 class: 33.4 ± 3.6 h, *p <* 0.001). Similarly, in the 21–30 age class, mated females fed the restricted diet lived longer compared to those in the 31–40 age class (mated females in the 21–30 class: 70.9 ± 6.9 h; in the 31–40 age class: 42.3 ± 2.7 h; *p* = 0.001) and also to those in the 41–50 age class (in the 41–50 class: 33.4 ± 3.6 h, *p* < 0.001) (Figure 7B, Appendix A).

In all four age classes, no significant differences in starvation resistance were observed between virgin and mated females that were fed the full diet (Figure 7A). In some age classes, virgin females fed the restricted diet were more resistant than mated ones. Specifically, in the 11–20 age class, virgin females fed the restricted diet were more resistant than mated females (virgins: 107.8 ± 3.7 h; mated: 78.9 ± 7.8 h; *p* = 0.012). In the 31–40 age class, virgin females fed the restricted diet were more resistant to starvation than mated females (virgins: 69.0 ± 4.8 h; mated: 42.3 ± 2.7 h; *p* = 0.002), and in the 41–50 age class, virgin females fed the restricted diet were also more resistant to starvation than mated ones (virgins: 88.3 ± 7.1 h; mated: 33.4 ± 3.6 h; *p* < 0.001) (Figure 7B).

In the 11–20 age class, virgin females fed the full diet had reduced resistance to starvation compared to those fed sugar (virgins fed the full diet: 74.4 ± 2.4 h; virgins fed the restricted diet—sugar: 107.8 ± 3.7 h; *p* < 0.001). Similarly, in the 41–50 age class, virgin females fed the full diet exhibited reduced resistance to starvation compared to those fed sugar (virgins fed the full diet: 46.5 ± 1.5 h; virgins fed the restricted diet: 88.3 ± 7.1 h; *p* = 0.05). Additionally, in the 21–30 age class, mated females fed the full diet had decreased resistance to starvation compared to their sugar-fed counterparts (mated and fed the full diet: 43.4 ± 2.2 h; mated and fed the restricted diet: 70.9 ± 6.9 h, *p* = 0.001) (Figure 7A,B).

### 3.3. Percentage Starvation Resistance Differences from the Corresponding Gender Mean

In Figure 8, the starvation resistance of males and females was also calculated as the percent difference from the corresponding mean of each gender (Equation (1)). Based on these results, the trend identified for both genders regarding starvation resistance is almost the same, meaning that gender is not a factor that affects the longevity of the insects under stress conditions. In both genders and for all age classes, the virgin adults treated with a restricted diet (sugar) had a higher starvation resistance (for males: from 10.3 up to 46.5%; for females: 12.6 up to 75.9%) compared to the corresponding (male, female) mean. Also, in younger age classes (11–20 and 21–30), in both genders, virgin adults treated with the full diet (protein) (for males: from 28.4 up to 46.0%; for females: 3.9 up to 21.4%) and mated adults treated with the restricted diet (sugar) (for males: from 8.1 up to 33.5%; for females: 15.7 up to 28.7%) also had higher starvation resistance compared to the corresponding (male, female) mean. Conversely, lower starvation resistance was similar for both genders when compared to their (male, female) mean. In both genders and for all age classes, the mated adults treated with the full diet (protein) showed lower starvation resistance (for males: from −16.8 up to −49.1%; for females: −6.4 up to −47.8%) compared to the corresponding mean of their gender. Also, in older age classes (31–40 and 41–50), in both genders, virgin adults treated with the full diet (protein) (for males: from −1.6 up to −19.7%; for females: −13.1 up to −24.1%) and mated adults treated with the restricted diet (sugar) (for males: from −16.6 up to −31.4%; for females: −31.0 up to −45.5%) also had lower starvation resistance compared to their (male, female) mean.

In sum, the results show that (a) resistance to food deprivation decreases with age, (b) in each age class, virgins are more resistant than the corresponding mated flies, (c) in each age class, adults on the restricted diet are more resistant than those fed a full diet. Mean values also indicate that females withstand food deprivation more than males in almost all treatments.

## 4. Discussion

The results of our study revealed crucial insights into the conditions of age, diet, and mating status, under which *B. oleae* is more susceptible to food deprivation. These findings can lead to the implementation of more effective and environmentally sound pest management strategies that focus on the conditions under which *B. oleae* is more vulnerable. These findings can not only provide critical information for more efficient pest control strategies but also improve the understanding of the physiology of the insect to develop integrated pest management strategies that focus on the weaknesses of *B. oleae*. In this way, new, effective, and sustainable strategies can be developed that are not harmful to beneficial insects, human health, or the environment. This new sustainable approach would be of great interest to olive-growing countries that are constantly seeking the formulation of more effective pest control strategies.

To our knowledge, our work is the first and unique study in the starvation scientific literature that investigates how starvation resistance in the oligophagous insect *B. oleae* is influenced by aging (across each day and for the age range from the 11th to the 50th day). Due to the lack of other similar studies, there is difficulty in the assessment of our findings. The only study that is close to the topic of our work but concerning a different insect from the same family (Tephritidae) is one that was carried out and published by our laboratory on the cosmopolitan and polyphagous species *C. capitata,* in which it was examined how resistance to food deprivation changes across each day and throughout the entire adult lifespan of the insect [16]. Overall, our findings are like those on *C. capitata*; they specifically indicate that starvation resistance declines with age in both genders, and younger adults endure longer. In addition, our results on the effect of diet on starvation resistance in *B. oleae* are also in line with the results from the study for *C. capitata*; adults fed the restricted diet show a higher starvation resistance than those fed the full diet. In our work, olive fruit flies fed the restricted diet show a higher starvation resistance in the first age class (11–20 days), followed by an abrupt decline in the second age class (21–30 days) and, finally, a small increase in the other two older age classes (31–40 and 41–50 days), which, as a result, is in line with the results of the study on *C. capitata* [16]. A comparison with other studies is not considered feasible due to the heterogeneity of the studies (i.e., with studies on *D. melanogaster* or other insects that belong to the same order but to different families). However, due to the lack of similar studies, a comparison will be attempted in the same order-based context to achieve some sort of assessment. Moreover, virgin adults of *D. melanogaster* fed a full diet had—in relative terms—a lower resistance to starvation [48]. Furthermore, *D. melanogaster* adults fed a restricted diet showed the reverse pattern of starvation resistance, with age positively affecting resistance in females and without any effects on males [46].

Experiments with wild *B. oleae* are inherently difficult to conduct, as this insect requires fresh olive fruit to complete its life cycle. As a result, olives must be uninfested and should be harvested in time to effectively rear olive fruit flies. These olives can be preserved in scientific refrigerators but for no longer than 4 months, setting a barrier to the completion of the experiments. Also, it is difficult to obtain large numbers of insects simultaneously for a given period. To maintain the number of insects needed for the experiment, infested olives were constantly collected from olive groves for a period of approximately three months. In addition, specific constant laboratory conditions are required for rearing and for conducting experiments with *B. oleae* adults, such as temperature (25 ± 2 °C), relative humidity (65 ± 5%), and photoperiod (L14:D10).

Moreover, it must be stressed that the insects cannot be easily synchronized; this means that to obtain the number of insects for each treatment, at least 2–3 days were needed for younger insects, and at least 20–25 were required for the older ones. In addition, not all insects underwent starvation simultaneously. For each day from the 11th to the 50th day of their lifespan, 80 insects per treatment were subjected to starvation, increasing the complexity of the experiment and the difficulty in the management of the measurements. The pattern of subjecting insects to starvation daily is based on a similar approach adopted from a previous study on Tephritidae [16]. Therefore, conducting experiments with wild *B. oleae* is a challenging process, and this might be the reason why no similar experiments have been conducted on starvation resistance for this species so far.

During the stage of collecting non-infested and infested olives, an effort was made to obtain olives and flies located in the region of Chalkidiki and Northern Greece to have more representative samples. To reduce the subjective bias errors in the measurements, the same person from our scientific team was responsible for recording insect deaths due to starvation. In addition, the rotation of the individual cages (change in their position in the laboratory on a circular basis) was performed daily to reduce potential experimental errors. As a result, all of these efforts to reduce biased errors further increased the complexity of the experiment and the time needed for its completion.

**Mating-specific effects on starvation resistance**. In our study, virgin adults exhibited greater starvation resistance compared to mated adults in all age classes. The results of our study show that mating can be energetically costly for both males and females, and as a result, virgin adults exhibit higher starvation resistance compared to mated ones. These results are in line with previous studies on insect mating status [24,49,50]. Male *C. capitata* that courted but did not mate have a similar lifespan to those that courted and then mated, meaning that courtship alone is metabolically costly [50]. It was also found that mating decreased the locomotor activity levels of males; these findings provide evidence that both courting and mating are metabolically costly [36]. Males may need to compete for access to mates or invest in courtship displays. Wing vibration associated with courtship [51,52] and the spreading of pheromones [53] result in a shorter lifespan for males [50]. Moreover, energetically expensive traits such as wings with special morphological characteristics that signal male quality and attractiveness to potential mates are costly to produce and maintain [51,52]. In *Bactrocera tryoni* (Froggatt) (Diptera: Tephritidae), mated females seem to be less resistant to starvation than virgins, and that may happen because of mating attempts and due to the mating procedure [54]. These mating procedures cost more energy in mated females than virgins, the latter being more resistant to starvation. A recent study showed that in two different strains of *Anastrepha ludens* (Loew) (Diptera: Tephritidae), sexual maturation reduced survival in both genders [55].

**Age-specific effects on starvation resistance**. The results of the present study show that younger individuals exhibit higher starvation resistance than those in older age classes. This can be explained by the fact that younger adults may have greater energy reserves from the larval stages than older ones or more efficient metabolic processes that allow them to better endure periods of food deprivation. Gerofotis et al., following a similar methodology to that used in the present study, found that in *C. capitata*, starvation resistance declines with age; age and adult diet were the most significant determinants of starvation resistance, followed by gender [16]. Belyi et al. observed a strong negative correlation between age and starvation resistance in *D. melanogaster* [56]. Experiments on *B. tryoni* showed that resistance to starvation and desiccation in both males and females decreases with age, although there were statistically significant differences in the pattern and extent of this decline [57]. Throughout aging, other biological functions also decrease. In *B. tryoni*, the effect of age on the olfactory response and exploratory activity was found to be important; specifically, the probability of an olfactory response in both genders to test odors declined with age [28].

**Diet-specific effects on starvation resistance**. Our experiments revealed a clear relationship between diet and starvation resistance across all age classes, and specifically, adults fed a protein-rich diet displayed a lower resistance to starvation compared to adults fed a sugar-rich diet. Dietary restriction has already been studied in many species. Experiments on *A. ludens* showed that a restricted diet not only extended their longevity but also reduced their reproductive output [58]. Similarly, in females of *Anastrepha fraterculus* (Wiedemann) (Diptera: Tephritidae), protein restriction expanded longevity and decreased egg production [59]. Also, in *B. tryoni*, lifespan and egg production were often closely linked to diet and the consumption of micronutrients [13]. In addition, protein consumption accelerates sexual maturation, leading to further energy losses due to mating and, as a result, to a shorter lifespan [21,22,23,24]. In the case of sugar consumption, the female produces a smaller number of eggs and, therefore, expends less energy [16,60]. These patterns are in line with our findings and explain the higher starvation resistance of virgin adults when fed a restricted diet. In females, the consumption of a protein diet plays an important role in the maturation of their eggs. Specifically, in *B. tryoni* this laboratory adaptation has been found to significantly enhance fecundity, resulting in a notable 4- to 5-fold rise in the rate at which dietary protein is converted into eggs; mated females seem to be less resistant to starvation than virgin ones on different diets, and that may result from mating attempts and due to the mating procedure [54]. In the case of *C. capitata*, it has been observed that mating enhances egg production in protein-fed females, but this is counterbalanced by reduced survival [61]. In *Anastrepha serpentina* (Wiedemann) (Diptera: Tephritidae), egg production was highest in flies maintained on a protein-rich diet; in flies kept on a restricted diet, egg production was lower [37]. It has been found that in *B. oleae*, mating negatively affects female longevity [62].

In Tephritid flies, protein digestion can be more energetically expensive than sugar digestion, leaving fewer resources for other essential functions, such as maintaining energy reserves [10,11,63,64,65]. Specifically, in *C. capitata*, nutritional status is a dominant factor known to affect the male’s signaling performance and determine the female’s decision to accept a male as a sexual partner [63]. Mating and a full diet (protein) shorten longevity in both genders. In *C. capitata*, wild males fed protein had a mating advantage over protein-deprived males [64]. These activities can require significant energy expenditure, leaving fewer resources available for essential functions, such as maintaining energy reserves, thus leading to shorter lifespans [10,11,65]. Research on *D. melanogaster* indicated that a seminal fluid protein in stored sperm, the molecule Acp26Aa, is responsible for an initial elevation in egg laying; females mated to mutant males that lack the molecule Acp26Aa lay fewer eggs than those mated with wild males [66].

A recent study on *Bactrocera dorsalis* (Hendel) (Diptera: Tephritidae) showed that a restricted diet leads to changes in phenotype, antioxidant response, and gene expression and a prolongation of lifespan in this species [67]. It would be interesting to study whether *B. oleae* may experience similar negative effects when fed a restricted diet. *Anastrepha ludens* and *Anastrepha obliqua* (Macquart) (Diptera: Tephritidae) fruit flies exposed to a combination of sugar and fresh mango fruit pulp showed greater longevity and field survival and better mating performance [38]. Experiments on *B. oleae* with a diet based on fresh fruits, their derivatives, or fruit pulps may lead to similarly useful insights. According to our findings, insects fed a full diet exhibited reduced starvation resistance in laboratory conditions. Therefore, in future experiments on *B. oleae*, it is suggested that fruit-based diets be preferred to protein to examine field survival and mating performance while limiting the potential negative effects of protein on starvation resistance that have been identified under laboratory conditions. These results could be applied to sustainable and environmentally friendly control methods for *B. oleae*, such as the sterile insect technique or other biotechnological methods.

**Three-factor interactions in starvation resistance**. In a study on *C. capitata*, it was found that as the cohort aged, there was a noticeable trend of smaller consecutive lipid crests, and as the cohort reached advanced ages and approached the maximum age, the lipid contents experienced a significant decline [68]. These findings are in line with our results: between the ages of 11 and 20 days, mated males that had consumed a full diet exhibited a lower resistance to starvation. As mated adults reached the age of 21 to 30 days, differences in starvation resistance were observed between those that had consumed full or restricted food. This variation arises from the ongoing maturation of their eggs, as well as the influence of courtship, competition, and mating in both genders. However, based on our results, after reaching 30 days of age, the diet consumed does not appear to affect starvation resistance in mated adults. Conversely, in the age group of 41 to 50 days, there is a discrepancy in starvation resistance between virgin males and females, but no significant difference is observed among mated individuals. This suggests that a restricted diet enhances the resistance of adults in this age range when it comes to starvation resistance.

Further research should focus on the isolation and comprehensive investigation of genes associated with fertility, development, and ovarian function. For instance, a study conducted on *Zeugodacus cucurbitae* (Coquillett) (Diptera: Tephritidae) provided genetic insights into the intricate processes underlying ovarian development and reproductive outcomes affected by nutritional factors [69]. The tangible next steps in *B. oleae* research will first be studies in natural olive grove conditions, where the influence of factors affecting insect starvation resistance may vary. Secondly, studying lipid contents in *B. oleae* could provide additional useful insights into the proximal physiological processes underlying starvation resistance.

Our results have strong practical implications for pest control. The study of adult stress resistance in insects could ultimately lead to (early-life) rearing protocols that would enhance the physiological traits of sterile mass-reared males (e.g., starvation and desiccation resistance). Based on the results of our study, we recommend further research on the use of sugar with fruit derivatives as a diet for mass-reared males (which is also more cost-effective compared to yeast hydrolysate diets) because it could boost starvation resistance and conceivably other fitness traits, ultimately benefiting the efficacy of the sterile insect technique. The same applies to other “genetic” methods where the release of mass-reared insects is required. The findings of the current study can therefore be used to formulate more effective strategies based on better knowledge of the insect’s biology. Following up on our findings, additional studies should seek to investigate the physiological mechanisms behind starvation resistance, thus providing essential benefits for the management of species of insects of agricultural and medical importance.

## 5. Conclusions

The results of the current study reveal insights regarding the key factors affecting starvation resistance in *B. oleae*: mating status, age, and diet. This information is important for improving existing or formulating new, more effective, and environmentally sound pest control strategies.

The main conclusions of the current study regarding the factors affecting starvation resistance are the following:*Mating status*: virgin adults exhibit higher starvation resistance compared to mated adults in all age classes.*Age*: younger adults exhibit higher starvation resistance in almost all treatments.*Diet*: adults that are fed a full diet containing protein show notably lower starvation resistance compared to sugar-fed ones.*Gender*: the same pattern of starvation resistance has been identified for both genders.

Regarding the interactions between the above factors, it can be deduced that in both genders, younger virgin adults fed the restricted diet show higher resistance in conditions of food deprivation.

We expect that the findings from our study on the critical factors of age, diet, and mating status that affect starvation resistance in *B. oleae* will provide valuable information on the vulnerability of this insect to food deprivation. Field studies and further research must be conducted to confirm the results of our study on a larger scale. 

## Figures and Tables

**Figure 1 insects-14-00841-f001:**
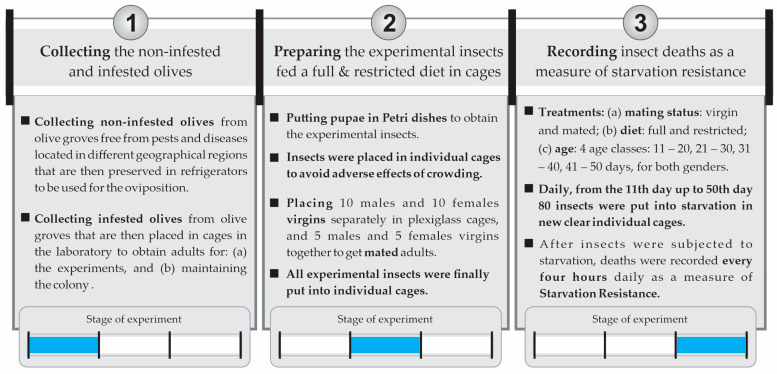
Stages of the experiment: (1) collecting non-infested and infested olives; (2) preparing experimental insects fed full (protein) and restricted (sugar) diets; (3) recording insect deaths as a measure of starvation resistance.

**Figure 2 insects-14-00841-f002:**
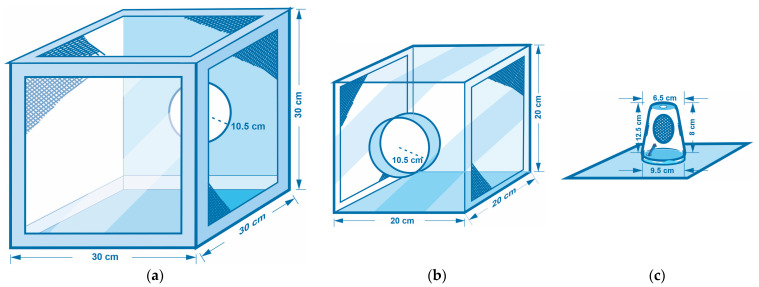
Blueprints of the experimental cages used for fly rearing under constant laboratory conditions: (**a**) BugDorm-type (custom-made) cages for rearing; (**b**) transparent plexiglass cages to which flies were transferred (on the 10th day) to mate or to be together with others of the same sex; (**c**) plastic cages (also custom-made from plastic cups) as individual cages.

**Figure 3 insects-14-00841-f003:**
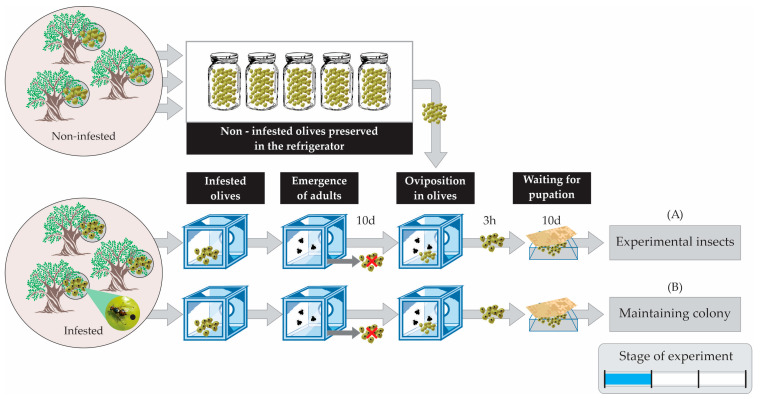
Collecting healthy and infested olives to rear (A) the experimental insects and (B) flies for maintaining the colony.

**Figure 4 insects-14-00841-f004:**
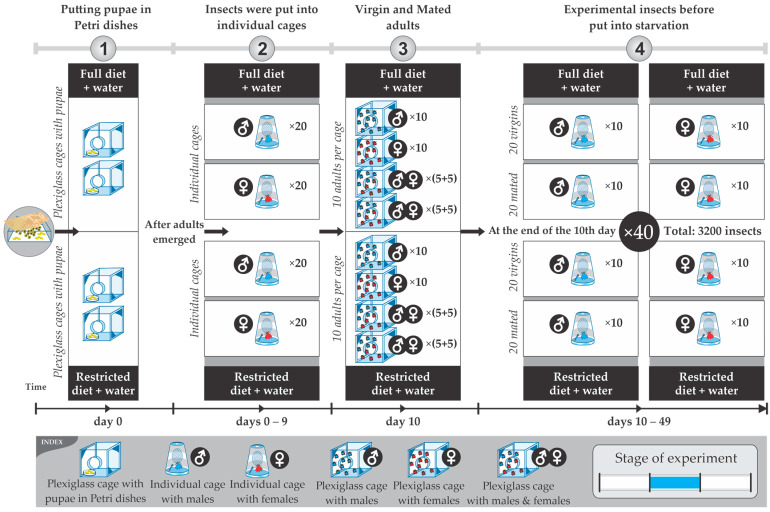
Schematic representation of the processes for the preparation of the experimental insects before they underwent starvation: (1) pupae were put in Petri dishes in plexiglass cages; (2) after adults emerged, insects were put into individual plastic cages to avoid crowding; (3) on the 10th day, all insects were transferred to 8 plexiglass cages: 4 cages with full diet (40 adults in total: 1 cage with 10 males, 1 with 10 females, and 2 cages with 5 males and 5 females in each cage); (4) at the end of the 10th day, insects were transferred back to their individual cages.

**Figure 5 insects-14-00841-f005:**
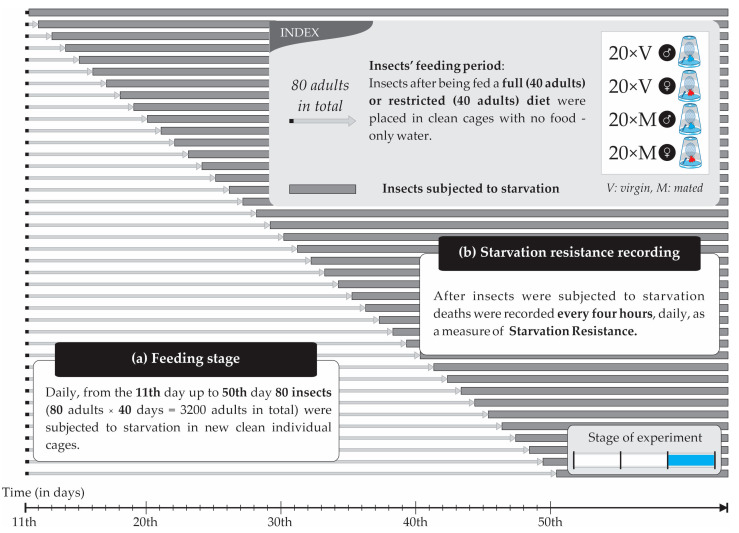
Schematic representation of the experimental design: (a) feeding stage: from the 11th day up to the 50th day of their lifespan, 80 insects (3200 in total) fed with the full or restricted diet were subjected to starvation in new clear individual cages; (b) starvation resistance recording: within the period of the 11th to 50th day, deaths were recorded every 4 h.

**Figure 6 insects-14-00841-f006:**
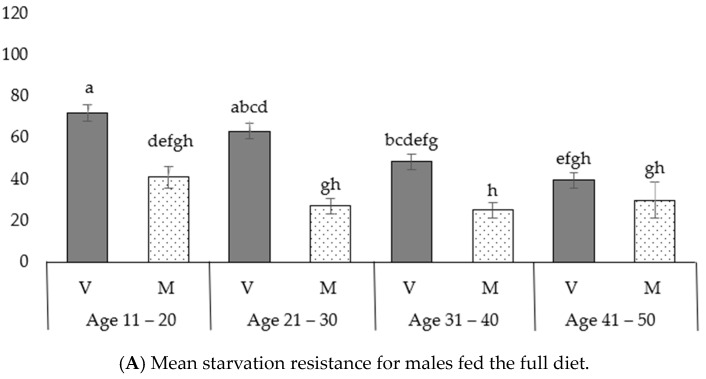
Starvation resistance (in hours to death) for males: (**A**) fed the full diet (protein); (**B**) fed the restricted diet (sugar). For both figures (**A**,**B**), bars with different lower-case letters above them correspond to differences in mean values that are statistically significant, at a significance level of *a* = 0.05, according to the results of Tukey’s test. Error bars correspond to standard errors of the mean. All 16 mean values (16 treatments: 2 diets × 4 age classes × 2 mated statuses) are comparable across the two figure sections.

**Figure 7 insects-14-00841-f007:**
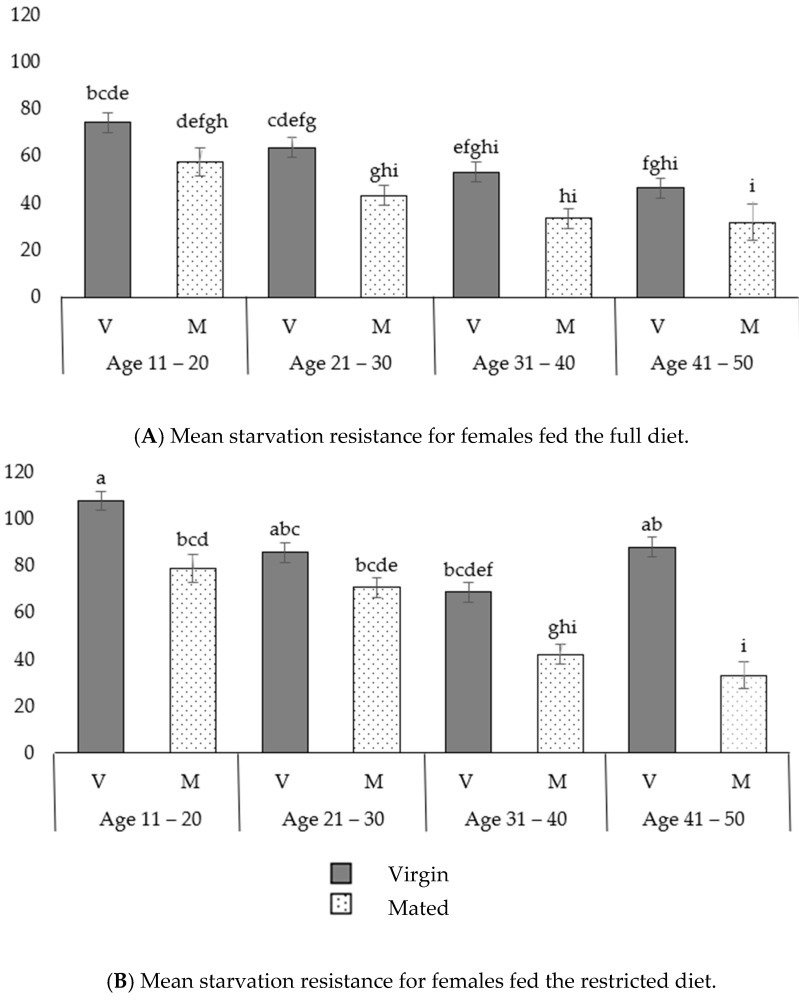
Starvation resistance (in hours to death) for females: (**A**) fed the full diet (protein); (**B**) fed the restricted diet (sugar). For both figures (**A**,**B**), bars with different lower-case letters above them correspond to differences in mean values that are statistically significant, at a significance level of *a* = 0.05, according to the results of Tukey’s test. Error bars correspond to the standard errors of the mean values. All 16 mean values (16 treatments: 2 diets × 4 age classes × 2 mated statuses) are comparable across the two figure sections.

**Figure 8 insects-14-00841-f008:**
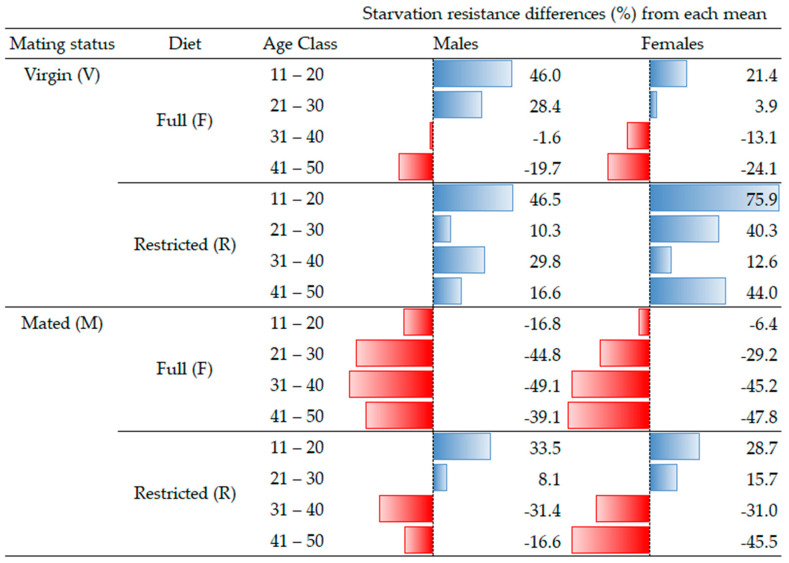
Percentage starvation resistance differences for males and females from the corresponding gender mean.

## Data Availability

Data will be available upon reasonable request.

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
