# Peer review of "The Roles of Mating, Age, and Diet in Starvation Resistance in Bactrocera oleae (Olive Fruit Fly)"

_insects, 2023, doi:10.3390/insects14110841_

Round 1

Reviewer 1 Report (Previous Reviewer 2)

The authors have addressed all my comments and I think that the manuscript can be published. However, I still have some minor corrections.

Generally, there are a lot of small mistakes and misprints in the text and it is very possible that I have missed some of them. Dear authors, please, carefully check the manuscript.

Line 93: Consider replacing “Diptera (Tephritidae)” either with “Tephritidae (Diptera)” or with “Diptera: Tephritidae”

Line 117: Do not capitalize “Melanogaster”.

Figure 1: Replace in the 2nd column “in petri” with “in Petri dishes”.

Figure 2a is not cited in the text (supposedly, it should be cited in lines 158-160).

Line 240:  replace “were placed in petri” with “were placed in Petri dishes”.

Figure 4: Replace “in petri” with “in Petri dishes” (twice!).

Line 253:  replace “in petri” with “in Petri dishes”.

Lines 283-289: Supplementary tables (as well as the tables in the main text) should be numbered in accordance with their citation in the text. Therefore table S5 (line 283) should not be the first cited.

Figures 6 and 7: replace “mean of starvation resistance” with “mean starvation resistance”.

Line 353: replace “are statistically significant different” with “are statistically significantly different”.

Line 400: replace "shew reduced" with "show reduced".

Line 615: delete comma after “et al”.

Author Response

Response to reviewer #1

Comment #1: The authors have addressed all my comments and I think that the manuscript can be published. However, I still have some minor corrections.

Response to comment #1: We would like to thank the reviewer for the valuable feedback and the time spent on the improvement of our work. We have addressed all minor corrections suggested.

Comment #2: Generally, there are a lot of small mistakes and misprints in the text, and it is very possible that I have missed some of them. Dear authors, please, carefully check the manuscript.

Response to comment #2: Thank you for this comment. We have carefully proofread the manuscript, and we made all the modifications and corrections needed.

Comment #3:

Line 93: Consider replacing “Diptera (Tephritidae)” either with “Tephritidae (Diptera)” or with “Diptera: Tephritidae”.

Response to comment #3: Done. We made an improvement.

Comment #4: Line 117: Do not capitalize “Melanogaster”.

Response to comment #4: Done. We did the correction.

Comment #5: Figure 1: Replace in the 2nd column “in petri” with “in Petri dishes”.

Response to comment #5: Done. We did the replacement as suggested.

Comment #6: Figure 2a is not cited in the text (supposedly, it should be cited in lines 158-160).

Response to comment #6: Done. Unfortunately, there was an issue with the performance of the Mendeley citation app, so actually the citation for figure 2a was originally there but was removed after Mendeley did an update on citations. Now, we believe that it is ok.

Comment #7: Line 240:  replace “were placed in petri” with “were placed in Petri dishes”.

Response to comment #7: Done. We did the replacement as suggested.

Comment #8: Figure 4: Replace “in petri” with “in Petri dishes” (twice!).

Response to comment #8: Done. We did the replacement as suggested.

Comment #9: Line 253:  replace “in petri” with “in Petri dishes”.

Response to comment #9: Done. We did the replacement as suggested.

Comment #10: Lines 283-289: Supplementary tables (as well as the tables in the main text) should be numbered in accordance with their citation in the text. Therefore, table S5 (line 283) should not be the first cited.

Response to comment #10: Done. Thank you for this comment. We added a short text that describes what the supplementary tables present and so supplementary tables are now numbered in accordance with their citation in the text.

Comment #11: Figures 6 and 7: replace “mean of starvation resistance” with “mean starvation resistance”.

Response to comment #11: Done. We did the replacement as suggested.

Comment #12: Line 353: replace “are statistically significant different” with “are statistically significantly different”.

Response to comment #12: Done. We made an improvement.

Comment #13: Line 400: replace "shew reduced" with "show reduced".

Response to comment #13: Done. We did the replacement as suggested.

Comment #14: Line 615: delete comma after “et al”.

Response to comment #14: Originally, we did the correction but just after following the suggestion of another reviewer we finally replaced “Nestle et al” with the phrase “In a study on C. capitata, it was found” and we added this citation at the end of the sentence. We believe that it is ok now.

Reviewer 2 Report (Previous Reviewer 3)

The manuscript insects-2628724 is a resubmission of a manuscript that was previously rejected.

General comments:

I want to start by acknowledging the improvement in the manuscript regarding the explanation of the experiments.  The new figures are very informative on the experimental design and protocol authors followed in their research.  I think these new figures are a great addition to the manuscript because they will allow readers to clearly appreciate how the research was conducted and the dimension and structure of the experiment.  Congratulations to the authors on this!  A suggestion to further improve the appreciation of the methods followed in the research is to present Figures 3, 4 and 5 as panels (a), (b) and (c) of a single figure.  For this, it will be necessary to increase the font size of the text in the figures so it can be totally readable.  This will improve appreciation of all stages of the research in a single figure instead of jumping from one figure to another.  Other specific comments on figures are provided in the list of comments below.

Overall, I feel that this revised manuscript improved a great deal in the graphical explanation of the experimental work.  On the other hand, the manuscript still needs improvement in its line of reasoning, especially in the introduction and discussion sections.  That is something I recommended in my previous review.  However, some of the specific recommendations I gave to the authors in my previous review were ignored without an explanation in the response to reviewer's letter as to why they were so.  For example, in my first general comment, I recommend that the authors ask a native English-speaking colleague to review their manuscript before submission.  But neither in the response to reviewers’ letter nor in the manuscript is there any mention acknowledging the colleague who reviewed the manuscript.  So, I assume that this specific recommendation was not considered by the authors.  Please explain why.

In my previous review I suggested authors the following structure for the Introduction: "(1) first paragraph, introduce the topic and highlight why it is important; (2) second paragraph, relate to current knowledge, what has been done and what needs to be done; (3) third paragraph, introduce your work clearly stating your objectives, and ideally hypothesis and predictions".  And the authors replied to this suggestion with: "Done. We would like to thank the reviewer for these valuable comments. The introduction has been rewritten following the suggestions of the reviewer".  But the authors did not really follow the recommended structure (the new introduction is structured into six paragraphs).  So, I do not understand why the authors say that they followed my suggestions when they did not.  Please explain.

In their response to reviewers' letter authors indicate they added the following text to explain how the residuals were tested for normality and equal variance: "The residuals' normality assumption was examined by visual inspections...".  But the manuscript I reviewed does not include such a text.

Below I provide specific comments to help authors improve their manuscript.

Line 11-18, suggest the following style "... on a full diet (water:sugar:Yeast Hydrolysate as protein), or a restricted diet without protein only with sugar, for both males and females.  Mated adults showed much less starvation resistance compared to virgin adults.  Younger adults endured longer periods of starvation resistance compared to __.  Adults fed on the sugar-only diet endured starvation for longer periods than those fed on the full diet.  We conclude that __.".  Or find another way to present text in lines 11-18, because the way it is currently presented is hard to follow especially for a simple summary.

The abstract has many unnecessary texts hindering the flow of ideas and such text can be safely removed/modified.  For instance, the text that is currently presented in lines 21-28 can be summarized into the following text "... food deprivation.  In the present study, we examined the influence of the mating status (mated/virgin), the age class (indicate the age classes), and diet quality (protein plus sugar, or sugar only diet), on starvation resistance of B. oleae".  There is a repetition of text in the simple summary and abstract "Findings indicated that under the same...".  Authors need to make a better effort to present in a clear and succinct way their most important findings and conclusions.  Be aware that many readers will only read the abstract of your paper and you must make out the most of it.

Line 60, you never talked before about artificially reared insects.  So, it is not clear why you bring this up here.

In fact, the text in lines 59-63 has little meaning in the way it is presented.  I guess this text is formulated in the context of mass-rearing for SIT?  But it is not the role of readers to be guessing what the authors want to communicate.  Authors should communicate their ideas precisely and clearly leaving no room for interpretation.

Line 79, add citations for "... and ecology of insects".

Line 92-94, this sentence is misleading.  Starvation resistance has been studied in other tephritid species different from Ceratitis capitata. The authors even cite other works in their manuscript in which starvation resistance was studied with tephritid species (e.g., Utgés et al. (2013)).  Authors must recognize and cite previous studies on the topic of starvation resistance in tephritids.  They should make a deeper review of the literature on starvation resistance of tephritid flies.

Lines 96-97, revise and improve the clarity of the text "assessing the impact on olive production".

Line 137-147, this text could be moved to the part where you explain the experimental protocols/procedures; and start this section with "2.1 Origin and handling of experimental insects and olive fruits".

The text in Figure 1 needs editions, e.g., stage (2): in the first point: Petri dish; second point: do you mean "Insects were placed individually in cages to avoid adverse effects of crowding"?

I think Figure 2 would be better presented as supplemental material.

Line 188, starvation resistance in hours/minutes...?  Indicate units.

Line 190-191, Wouldn't the experimental unit be the individual flies?

Line 199-200, explain what a replicate is.

Line 204-206, this text does not contribute to anything here.  Can be safely removed.

In Figure 5, I could not find (a) and (b).  Most of the text in axes or legends is too small and does not read well.  Increase the font size to make it readable.

In the Statistical analyses section, you should include a mention of how the assumptions of normal distribution and equal variance of the residuals were checked.  Also, indicate if you used a post-hoc test.  Cite the statistical software used.

Line 284-288, It is not clear why authors focused on using the "simple-simple effects analysis" as a consequence of finding significant three-way interaction effects of the predictors in starvation resistance of flies.  Explain.

I did not understand what, exactly, figure 8 is showing.  Authors need to improve their explanation of how to interpret this figure.  The description/explanation of the analysis associated with this figure needs to be included in section 2.4 Statistical analyses (with an "e" not an "i").

The first paragraphs of the discussion (lines 463-497) explain many of the difficulties that the authors encountered in conducting the study with B. oleae, and give methodological details of the efforts made to avoid errors in data collection.  I agree with this text being included in the discussion.  However, I do not think that the beginning of the discussion is the best place for this text.  The discussion should begin by highlighting the most important results of the research and what they mean in the context of the study.  Starting the discussion by explaining methodological difficulties is not a good strategy because it takes away authority from the authors' voice.  You should start the discussion by highlighting your most important findings with an authoritative voice.

Lines 464-467, how, exactly, the study of starvation resistance can be applied to "update or develop integrated pest management strategies".

Lines 587-589, cite references to support this statement.

Line 615, revise "Nestel et al" and include it in the reference list.

Lines 659-661, how, exactly, "a better understanding of the insects’ biology", "help improve environmentally friendly methods such as the Sterile Insect Technique and other genetic methods"?  You need to be specific and provide strong and deep arguments.

Line 676-677, how do you propose that your results be useful to "formulate more sustainable and effective control methods"?

Authors need to make their manuscript reviewed by a native English speaker prior to submission.

Author Response

Response to reviewer #2

We would like to thank the reviewer for the careful and insightful review of our manuscript and for the valuable comments, effort and time spent to improve the quality of our manuscript. Based on reviewer comments and suggestions, we have made several modifications to the original manuscript, and we also rewrote some parts of the text. After the changes applied, a native speaker has carefully proofread the manuscript to minimize all kind of linguistic errors regarding English. In our response to reviewer text, a point-by-point detailed description is given of how each comment has been addressed and what improvements have been made throughout the manuscript.

Comment #1: I want to start by acknowledging the improvement in the manuscript regarding the explanation of the experiments.  The new figures are very informative on the experimental design and protocol authors followed in their research.  I think these new figures are a great addition to the manuscript because they will allow readers to clearly appreciate how the research was conducted and the dimension, and structure of the experiment.  Congratulations to the authors on this!  A suggestion to further improve the appreciation of the methods followed in the research is to present Figures 3, 4 and 5 as panels (a), (b) and (c) of a single figure.  For this, it will be necessary to increase the font size of the text in the figures so it can be totally readable.  This will improve appreciation of all stages of the research in a single figure instead of jumping from one figure to another.  Other specific comments on figures are provided in the list of comments below.

Response to comment #1: We would like to thank the reviewer for the kind words on our effort to clearly describe and present the experimental processes both in the text and via the added figures. We appreciate the reviewer’s valuable comments, and we thank her/him for the suggestions. We applied several changes to further improve the readability of the figures such as enlarging the fonts and some minor position improvements in the graphics. We tried to place figures 3, 4, and 5 in a single figure but it is difficult to fit them all in a single page. For this reason, we moved the three figures and placed them as successive images, being as close as possible to each other giving the impression of a single figure. Regarding figure 4, we enlarged the graphics to match the width of the page.

Comment #2: Overall, I feel that this revised manuscript improved a great deal in the graphical explanation of the experimental work.  On the other hand, the manuscript still needs improvement in its line of reasoning, especially in the introduction and discussion sections.  That is something I recommended in my previous review.  However, some of the specific recommendations I gave to the authors in my previous review were ignored without an explanation in the response to reviewer's letter as to why they were so.  For example, in my first general comment, I recommend that the authors ask a native English-speaking colleague to review their manuscript before submission.  But neither in the response to reviewers’ letter nor in the manuscript is there any mention acknowledging the colleague who reviewed the manuscript.  So, I assume that this specific recommendation was not considered by the authors.  Please explain why.

Response to comment #2: Following the suggestion of the reviewer, we made some improvements regarding English, and we asked for the help of an experienced native speaker and partner of our laboratory to make corrections and improve the quality of the manuscript regarding the language used. We hope that the improvements made are sufficient.

Comment #3: In my previous review I suggested authors the following structure for the Introduction: "(1) first paragraph, introduce the topic and highlight why it is important; (2) second paragraph, relate to current knowledge, what has been done and what needs to be done; (3) third paragraph, introduce your work clearly stating your objectives, and ideally hypothesis and predictions".  And the authors replied to this suggestion with: "Done. We would like to thank the reviewer for these valuable comments. The introduction has been rewritten following the suggestions of the reviewer".  But the authors did not really follow the recommended structure (the new introduction is structured into six paragraphs).  So, I do not understand why the authors say that they followed my suggestions when they did not.  Please explain.

Response to comment #3: We followed the suggestions of the reviewer incorporating the existing text from the introduction. Following the suggestion of the reviewer we made new improvements in the introduction to match the structure proposed. The first paragraph as suggested is about the problem that needs to be addressed and the importance of the topic. The second paragraph present all the relative work and studies conducted on this topic. The last paragraph introduces the purpose and objectives of current study. Regarding the “hypothesis and predictions “, we added the following sentence: “We hope that studying the key factors affecting B. oleae under food deprivation will provide insights that will help to improve existing pest control management or even formulate new more sustainable and effective strategies.”

Comment #4: In their response to reviewers' letter authors indicate they added the following text to explain how the residuals were tested for normality and equal variance: "The residuals' normality assumption was examined by visual inspections...".  But the manuscript I reviewed does not include such a text.

Response to comment #4: Unfortunately, there was an issue when updating citations using the Mendeley app; without noticing it, some sentences that were close to citations were automatically removed by Mendeley. We carefully checked the text and placed back the missing sentences. We are sorry for the inconvenience.

Comment #5: Line 11-18, suggest the following style "... on a full diet (water:sugar:Yeast Hydrolysate as protein), or a restricted diet without protein only with sugar, for both males and females.  Mated adults showed much less starvation resistance compared to virgin adults.  Younger adults endured longer periods of starvation resistance compared to __.  Adults fed on the sugar-only diet endured starvation for longer periods than those fed on the full diet.  We conclude that __.".  Or find another way to present text in lines 11-18, because the way it is currently presented is hard to follow, especially for a simple summary.

Response to comment #5: Thank you for this comment. We made several modifications and improvements, trying to follow the suggestions.

Comment #6: The abstract has many unnecessary texts hindering the flow of ideas and such text can be safely removed/modified.  For instance, the text that is currently presented in lines 21-28 can be summarized into the following text "... food deprivation.  In the present study, we examined the influence of the mating status (mated/virgin), the age class (indicate the age classes), and diet quality (protein plus sugar, or sugar only diet), on starvation resistance of B. oleae".  There is a repetition of text in the simple summary and abstract "Findings indicated that under the same...".  Authors need to make a better effort to present in a clear and succinct way their most important findings and conclusions.  Be aware that many readers will only read the abstract of your paper and you must make out the most of it.

Response to comment #6: Thank you for this comment. We made several modifications and improvements, trying to follow the suggestions.

Comment #7: Line 60, you never talked before about artificially reared insects.  So, it is not clear why you bring this up here.

In fact, the text in lines 59-63 has little meaning in the way it is presented.  I guess this text is formulated in the context of mass-rearing for SIT?  But it is not the role of readers to be guessing what the authors want to communicate.  Authors should communicate their ideas precisely and clearly leaving no room for interpretation.

Response to comment #7: Thank you for this comment. We added the following text: “… such as the Sterile Insect Techniques (SIT) … in stress environmental conditions, …“ to clearly indicate that artificially reared male insects are used for SIT purposes.

Comment #8: Line 79, add citations for "... and ecology of insects".

Response to comment #8: Done.

Comment #9: Line 92-94, this sentence is misleading.  Starvation resistance has been studied in other tephritid species different from Ceratitis capitata. The authors even cite other works in their manuscript in which starvation resistance was studied with tephritid species (e.g., Utgés et al. (2013)).  Authors must recognize and cite previous studies on the topic of starvation resistance in tephritids.  They should make a deeper review of the literature on starvation resistance of tephritid flies.

Response to comment #9: Thank you for this comment. We made modifications and improved this part of the text.

Comment #10: Lines 96-97, revise and improve the clarity of the text "assessing the impact on olive production".

Response to comment #10: We deleted this part.

Comment #11: Line 137-147, this text could be moved to the part where you explain the experimental protocols/procedures; and start this section with "2.1 Origin and handling of experimental insects and olive fruits".

Response to comment #11: Done.

Comment #12: The text in Figure 1 needs editions, e.g., stage (2): in the first point: Petri dish; second point: do you mean "Insects were placed individually in cages to avoid adverse effects of crowding"?

Response to comment #12: Done. We followed the suggestion, and we made the corrections.

Comment #13: I think Figure 2 would be better presented as supplemental material.

Response to comment #13: We would agree with the reviewer on this comment but the main reason we included figure 2 in the text is because cages are used also in the other more complex figures and so, we believe that presenting these cage types separately with larger size in a figure first it helps readability and comprehension of the following figures.

Comment #14: Line 188, starvation resistance in hours/minutes...?  Indicate units.

Response to comment #14: Done.

Comment #15: Line 190-191, Wouldn't the experimental unit be the individual flies?

Response to comment #15: Yes, the experimental units were flies studied individually on the starvation resistance. In the text we added the following: “… that were studied individually on starvation resistance”.

Comment #16: Line 199-200, explain what a replicate is.

Response to comment #16: Thank you for this comment. Following the suggestion, we added the following text: "There were also 10 replicates, namely 10 insects of the same gender and same mating status, and at the same age (day of life), also fed on the same diet".

Comment #17: Line 204-206, this text does not contribute to anything here.  Can be safely removed.

Response to comment #17: We deleted this part of the text.

Comment #18: In Figure 5, I could not find (a) and (b).  Most of the text in axes or legends is too small and does not read well.  Increase the font size to make it readable.

Response to comment #18: Done. We added (a) and (b) in figure 5. We also enlarge fonts to improve readability. Thank you for this comment.

Comment #19: In the Statistical analyses section, you should include a mention of how the assumptions of normal distribution and equal variance of the residuals were checked.  Also, indicate if you used a post-hoc test.  Cite the statistical software used.

Response to comment #19: Done. Unfortunately, there was an issue with the performance of the Mendeley citation app, this information was removed after Mendeley did an update on citations. Now, we believe that it is ok.

Comment #20: Line 284-288, It is not clear why authors focused on using the "simple-simple effects analysis" as a consequence of finding significant three-way interaction effects of the predictors in starvation resistance of flies.  Explain.

Response to comment #20: Done. Same as above, we believe now that the issue has been addressed.

Comment #21: I did not understand what, exactly, figure 8 is showing.  Authors need to improve their explanation of how to interpret this figure. The description/explanation of the analysis associated with this figure needs to be included in section 2.4 Statistical analyses (with an "e" not an "i").

Response to comment #21: Done. Thank you for this comment. We added the formula used (the equation in 2.4) for the calculation of the starvation resistance as percentage-difference from the total mean of the corresponding mean of each gender.

Comment #22: The first paragraphs of the discussion (lines 463-497) explain many of the difficulties that the authors encountered in conducting the study with B. oleae, and give methodological details of the efforts made to avoid errors in data collection.  I agree with this text being included in the discussion.  However, I do not think that the beginning of the discussion is the best place for this text.  The discussion should begin by highlighting the most important results of the research and what they mean in the context of the study.  Starting the discussion by explaining methodological difficulties is not a good strategy because it takes away authority from the authors' voice.  You should start the discussion by highlighting your most important findings with an authoritative voice.

Response to comment #22: Thank you for this comment as it helped us to reform and improve the discussion section. This part regarding the experimental difficulties was moved lower in the discussion. Also, following the suggestion, the discussion now starts with the most important findings of our work.

Comment #23: Lines 464-467, how, exactly, the study of starvation resistance can be applied to "update or develop integrated pest management strategies".

Response to comment #23: Thank you for this comment. We added some text to make clear that the findings of our study provide insights about the combination of the key factors (age, diet, and mating status) in which B. oleae is potentially more vulnerable, and therefore can be used for updating or improving the pest management strategies.

Comment #24: Lines 587-589, cite references to support this statement.

Response to comment #24: Done.

Comment #25: Line 615, revise "Nestel et al" and include it in the reference list.

Response to comment #25: Done.

Comment #26: Lines 659-661, how, exactly, "a better understanding of the insects’ biology", "help improve environmentally friendly methods such as the Sterile Insect Technique and other genetic methods"?  You need to be specific and provide strong and deep arguments.

Response to comment #26: Done. This part of the text has been re-written.

Comment #27: Line 676-677, how do you propose that your results be useful to "formulate more sustainable and effective control methods"?

Response to comment #27: Done. This part of the text has been re-written.

Comment #28:  Comments on the Quality of English Language

Authors need to make their manuscript reviewed by a native English speaker prior to submission.

Response to comment #28: The manuscript has been thoroughly reviewed by a native speaker and several linguistic improvements have been made throughout the text.

Round 2

Reviewer 2 Report (Previous Reviewer 3)

I read the new version of the manuscript by Balampekou and colleagues and appreciate the responses of the authors to my previous comments.  Despite a clear improvement in the explanation of the experiments with the addition of new figures, and a well-prepared response to reviewer letter, please consider the following comments that go in the same line as mentioned before of improving precision and clarity.

Line 24-25, suggest rewriting to improve clarity, e.g., "Our results showed that starvation resistance of B. oleae did not differ significantly between females and males".

Line 21-22, delete "2 levels" and "4 levels", it is obvious.  Indicate units of age class.

Line 25-26, it is not clear why the information is ordered with (a), (b) and (c).  This presentation does not make sense.

The style of the Abstract of this manuscript fails to follow the structure recommended in the journal template as it lacks "the main conclusions or interpretations" of the research.  As such, the abstract fails to flush out the significance of the research.  Need to think and work on this thoroughly.

Line 39-41, Include citations to support this statement.

Line 43-47, The length of this sentence is excessive.  For the sake of clarity, revise and split in only one idea per sentence.

Line 55-57, Briefly, explain why and how "Studying the key factors regarding food deprivation that affect B. oleae can potentially provide insights that will help to formulate more sustainable and effective control methods" and cite reference.

Line 58, suggest "interest in exploring sustainable methods to control B. oleae such as..." and cite reference.

Line 59-61, the way this sentence is formulated seems odd.  While it is true that the success of the SIT is based on sterile males being able to compete sexually with wild males and achieve copulations with wild females in the field, the reality is that to achieve this, millions of sterile males must be released so that they can overwhelm the numbers of wild males and can copulate with wild females.  This is because wild insects are more sexually competitive than sterile ones.  So, it is not accurate to say that mass-rearing facilities produce insects that are more competitive than wild males.  Revise.

Line 62-63, starvation is inherently a situation of food deprivation, so what is the difference between starvation resistance and food deprivation?

Line 71-72, who's "fitness"?  Or do you mean "affect their fitness"?  Also, need to clarify if the diets mentioned are larval diets or adult diets.

Line 76, "effects" in place of "effect".

Lines 79-82, Indicate if the protein-based diet was for larvae or adults.

Line 85-87, suggest "but to our knowledge, the starvation resistance of B. oleae has not been studied yet".  Also, isn't it redundant to say "starvation resistance to food deprivation"?  As mentioned before, starvation resistance is inherently a situation of food deprivation.

Line 91, "effect of (adult or larval?) diet on several...".

Lines 99-100, 103-107, cite appropriate references.

Line 108-111, The way the aims are presented in this new text is unclear and imprecise.  To begin with, there are two objectives in this sentence, therefore, you should say "aims" (or objectives, in plural) instead of "aim", and formulate the sentence in the past tense (i.e., The objectives of the study were...).  The first objective "identify in which mating status, age class, and diet, B. oleae is more vulnerable" does not make clear what, exactly, is meant by "vulnerable". Furthermore, the variables "mating status", "age class" and "diet" are not defined so the way these variables are presented here is meaningless at this point.  It is not clear what age classes and diets you investigated.  The second objective, "study the effect of the interaction of these three factors on the starvation resistance of this pest", is better formulated, but because before it is not explained what the authors refer to with "mating status", "age class" and "diet", the value of studying the interaction among these variables is not fully understood here.

I am really struggling to understand the rationale and value of the analysis reported in lines 315-320 and in Figure 8.  In line 319 define treatment.  And explain the objective of this analysis to improve its appreciation.  When you say "total mean of each gender" do you refer to the grand mean of all flies in each gender regardless of their mating status, age class and diet treatment?

Figure 5 still needs to improve the font size of "x" and "y" axes.  In the case of the "y" axis, it is impossible to read the text inside squares, and the timeline on the "x" axis does not read well either.

Lines 498-499, instead of saying "The results of our study revealed crucial insights about how age, diet, and mating status affect the vulnerability of B. oleae under food deprivation", indicate, specifically, what those "insights" are and what they mean.

In lines 499-500, indicate the specific information that is useful for "the implementation of effective and environmentally sound pest management". 

Line 501, indicate what is the "critical information for more efficient pest control strategies...".

Acknowledge the colleague who reviewed the English of your manuscript.

Acknowledge the colleague who reviewed the English of your manuscript.

Author Response

Response to the Reviewer

I read the new version of the manuscript by Balampekou and colleagues and appreciate the responses of the authors to my previous comments.  Despite a clear improvement in the explanation of the experiments with the addition of new figures, and a well-prepared response to reviewer letter, please consider the following comments that go in the same line as mentioned before of improving precision and clarity.

Response: Thank you for the kind comments on our effort to improve the quality of the manuscript. Also, thank you for the suggestions to improve it further.

Comment #1: Line 24-25, suggest rewriting to improve clarity, e.g., "Our results showed that starvation resistance of B. oleae did not differ significantly between females and males".

Response to comment #1: Thank you for this comment. We followed the suggestion, and we replaced the sentence.

Comment #2: Line 21-22, delete "2 levels" and "4 levels", it is obvious.  Indicate units of age class.

Response to comment #2: Thank you for this comment. Done.

Comment #3: Line 25-26, it is not clear why the information is ordered with (a), (b) and (c).  This presentation does not make sense.

Response to comment #3: Thank you for this comment. We removed the ordering, following the suggestion.

Comment #4: The style of the Abstract of this manuscript fails to follow the structure recommended in the journal template as it lacks "the main conclusions or interpretations" of the research.  As such, the abstract fails to flush out the significance of the research.  Need to think and work on this thoroughly.

Response to comment #4: Thank you for this comment. We re-wrote the last sentence of the abstract starting with "The main conclusions of our study regarding mating status, age, and diet indicated that ..”.

Comment #5: Line 39-41, Include citations to support this statement.

Response to comment #5: Thank you for this comment. Done.

Comment #6: Line 43-47, The length of this sentence is excessive.  For the sake of clarity, revise and split in only one idea per sentence.

Response to comment #6: Thank you for this comment. Done.

Comment #7: Line 55-57, Briefly, explain why and how "Studying the key factors regarding food deprivation that affect B. oleae can potentially provide insights that will help to formulate more sustainable and effective control methods" and cite reference.

Response to comment #7: Thank you for this comment. We deleted the sentence because later in the text it is being presented better and in detail.

Comment #8: Line 58, suggest "interest in exploring sustainable methods to control B. oleae such as..." and cite reference.

Response to comment #8: Thank you for this comment. Just after the text “… such as the Sterile Insect Techniques (SIT)” we added a reference.

Comment #9: Line 59-61, the way this sentence is formulated seems odd.  While it is true that the success of the SIT is based on sterile males being able to compete sexually with wild males and achieve copulations with wild females in the field, the reality is that to achieve this, millions of sterile males must be released so that they can overwhelm the numbers of wild males and can copulate with wild females.  This is because wild insects are more sexually competitive than sterile ones.  So, it is not accurate to say that mass-rearing facilities produce insects that are more competitive than wild males.  Revise.

Response to comment #9: Thank you for the suggestion. We revised the sentence as follows: “The efficacy of the method relies in producing mass – reared male insects that when are released in the wild in large numbers, they are finally more competitive compared to wild males, and, also capable of enduring more in stress environmental conditions”.

Comment #10: Line 62-63, starvation is inherently a situation of food deprivation, so what is the difference between starvation resistance and food deprivation?

Response to comment #10: Thank you for this comment. We deleted this part of the text: “whereas the food deprivation” to be clearer.

Comment #11: Line 71-72, who's "fitness"?  Or do you mean "affect their fitness"?  Also, need to clarify if the diets mentioned are larval diets or adult diets.

Response to comment #11: Thank you for the comment. We added “their” as suggested. We also added “adult” before “insects” to improve clarity.

Comment #12: Line 76, "effects" in place of "effect".

Response to comment #12: Thank you for the comment. Now the sentence is as follows: “The effects of food deprivation have been studied …”

Comment #13: Lines 79-82, Indicate if the protein-based diet was for larvae or adults.

Response to comment #13: Thank you for the comment. We added “adults” after “feeding”.

Comment #14: Line 85-87, suggest "but to our knowledge, the starvation resistance of B. oleae has not been studied yet".  Also, isn't it redundant to say "starvation resistance to food deprivation"?  As mentioned before, starvation resistance is inherently a situation of food deprivation.

Response to comment #14: Thank you, we made the replacement as suggested. We also agree the “food deprivation” is redundant, so we deleted it.

Comment #15: Line 91, "effect of (adult or larval?) diet on several...".

Response to comment #15: Thank you, we added “adult” before “olive fruit fly”.

Comment #16: Lines 99-100, 103-107, cite appropriate references.

Response to comment #16: Thank you for this comment. Done.

Comment #17: Line 108-111, The way the aims are presented in this new text is unclear and imprecise.  To begin with, there are two objectives in this sentence, therefore, you should say "aims" (or objectives, in plural) instead of "aim", and formulate the sentence in the past tense (i.e., The objectives of the study were...).  The first objective "identify in which mating status, age class, and diet, B. oleae is more vulnerable" does not make clear what, exactly, is meant by "vulnerable". Furthermore, the variables "mating status", "age class" and "diet" are not defined so the way these variables are presented here is meaningless at this point.  It is not clear what age classes and diets you investigated.  The second objective, "study the effect of the interaction of these three factors on the starvation resistance of this pest", is better formulated, but because before it is not explained what the authors refer to with "mating status", "age class" and "diet", the value of studying the interaction among these variables is not fully understood here.

Response to comment #17: Thank you for this comment. We replaced “aim” with “aims”. We also added this part of the text: “namely more susceptible to several stresses,” to explain what we mean by using the term “vulnerable”. For the first objective to define the variables “mating status”, “age class”, and “diet”: we added “(virgin, mated)” after mating status, “(11 – 20, 21 – 30, 31 – 40, 41 – 50)” after age classes in days, and “(full, restricted)” after diet. We hope that the second objective is now clear after defining the factors we study. To clarify the term “interaction”, we added at the end of the sentence: “, namely the combination of the values of the three factors that resulted in increased insect vulnerability due to starvation".

Comment #18: I am really struggling to understand the rationale and value of the analysis reported in lines 315-320 and in Figure 8.  In line 319 define treatment.  And explain the objective of this analysis to improve its appreciation.  When you say "total mean of each gender" do you refer to the grand mean of all flies in each gender regardless of their mating status, age class and diet treatment?

Response to comment #18: Thank you for this comment. In our effort to find a way to present better the influence of each of the three factors to insect starvation resistance, we calculated the difference (in %) between the values (in days) in each treatment and the total mean of all flies in each gender. To define “treatments” we added the following text: “(16 treatments in total: (mating status: virgin, mated) x (age classes: 11 – 20, 21 – 30, 31 – 40, 41 – 50) x (diet: full, restricted diet),”. Yes, “total mean of each gender” refers to the “grand mean of all flies in each gender regardless of their mating status, age class and diet”. We also made some modifications to the text, as suggested.

Comment #19: Figure 5 still needs to improve the font size of "x" and "y" axes.  In the case of the "y" axis, it is impossible to read the text inside squares, and the timeline on the "x" axis does not read well either.

Response to comment #19: Thank you for this comment. We made several improvements on Figure 5.

Comment #20: Lines 498-499, instead of saying "The results of our study revealed crucial insights about how age, diet, and mating status affect the vulnerability of B. oleae under food deprivation", indicate, specifically, what those "insights" are and what they mean.

Response to comment #20: Thank you for this comment. We re-wrote the first sentence of the discussion as follows: “The results of our study revealed crucial insights into the conditions of age, diet, and mating status, under which B. oleae is more susceptible to food deprivation. These findings can lead to the implementation of more effective and environmentally sound pest management strategies that focus on the conditions under which the B. oleae is more vulnerable.”.

Comment #21: In lines 499-500, indicate the specific information that is useful for "the implementation of effective and environmentally sound pest management". 

Response to comment #21: Thank you for this comment. We re-wrote the whole sentence as mentioned above.

Comment #22: Line 501, indicate what is the "critical information for more efficient pest control strategies...".

Response to comment #22: Thank you for this comment. We added the following text: “… more effective and environmentally sound pest management strategies that focus on the conditions under which the B. oleae is more vulnerable.”

Comment #23: Acknowledge the colleague who reviewed the English of your manuscript.

Response to comment #23: Thank you for this comment. We added in acknowledgments the colleague who reviewed the English of our manuscript.

This manuscript is a resubmission of an earlier submission. The following is a list of the peer review reports and author responses from that submission.

Round 1

Reviewer 1 Report

I have attached my peer review of this manuscript. I hope that you will find this constructive and helpful. I really like the premise of the paper and it is very interesting work.

Review of Manuscript “Mating, age and diet modulate starvation resistance in the olive fruit fly”

Brief Summary:

In this manuscript, the authors investigate how age, diet, and environment can affect starvation resistance in the olive fruit fly. The olive fruit fly is a significant pest in the olive industry and there is a lot of research on sustainable methods to control their populations using genetic strategies such as the sterile insect technique. By studying how starvation resistance is impacted in the olive fruit fly, recommendations can be made to optimize longevity and reproductive capacity for engineered olive fruit flies under conditions in the wild.

Reviewer Comments:

1) Is the manuscript clear, relevant for the field and presented in a well-structured manner?

The manuscript is very relevant for the field with some exciting ideas and applications for rearing methods of sterile insects for optimal pest management. The overall results are promising, but the structure, documentation, and explanation needs improvement (see below)

2) Is the manuscript scientifically sound and is the experimental design appropriate to test the hypothesis?Are the manuscript’s results reproducible based on the details given in the methods section?

The premise and experimental design in general are scientifically sound. However, the manuscript results are not reproducible in its current form for the following reasons:

a) Lines 89: I could not see any mention of the source for the olive fruit flies used in these experiments. Were these olive fruit flies caught from wild populations, or have they been reared in the laboratory for multiple generations? If reared in the laboratory, were isogenic lines used? Drosophila melanogaster flies caught from populations in the wild have been shown to harbor distinct polymorphisms in their individual genomes that can substantially alter phenotypes        (Here is an example of such a collection: https://bdsc.indiana.edu/stocks/wt/dgrp.html). In the context of this study, it is important to control for genetic variation wherever possible since the paper seeks to apply these findings to optimize genetically engineered sterile olive fruit flies for pest management. I do not know the current state of olive fruit fly genetics in terms of strains and mutants, and this is a crucial detail for building confidence in the experimental results and conclusions and the limitations therein.

b) Lines 91-92: The sentence "Larvae have been reared..." is a bit confusing to read. Were the olive fruits collected in the wild stored in the refrigerator until needed to rear larvae, or were the larvae inside the fruits and kept at a cold temperature?

Also, was the adult diet provided as-is (in powder form), or was it mixed with a solidifying agent such as agar? The amount of protein/sugar in the full diet and sugar in the restricted diet were not quantified in terms of weight and amount given to each adult. These details are crucial because excessive amounts of protein and/or sugar can affect overall metabolism (diabetes, etc.).

c) Lines 92 to 94: The sentence that includes "or 5 males and 5 females (mated)" is confusing because it implies that they were already mated prior to being placed in larger cages, which contradicts the previous sentence that each adult was kept in individual cages prior to them. If all adults were virgins until being placed in these larger cases, it would be much clearer to state that 5 virgin males and 5 virgin females were allowed to mate in the larger cases for 3 to 5 days before being placed back into individual cases without food.

d) Lines 96-97: How were these flies monitored to ensure that successful mating occurred?

e) Lines 102-103: How was survival resistance recorded every 4 hours? If it was done completely manually, this suggests that there was always someone on site to measure survival every 4 hours for 3-4 days straight. This detail would make more sense if video or some type of imaging equipment was used to facilitate the assay during nighttime hours.

Summary: There needs to be more quantitative metrics in the materials and methods such that anyone who wishes to replicate these findings can do so. If there are limitations to how much text can be in the Materials and Methods, consider typing up a protocol and publishing it at protocol websites such as protocols.io and refer to the protocol in the Materials and Methods section.

3) Are the figures/tables/images/schemes appropriate? Do they properly show the data? Are they easy to interpret and understand? Is the data interpreted appropriately and consistently throughout the manuscript? Please include details regarding the statistical analysis or data acquired from specific databases.

The figures in the paper are composed solely of bar graphs and are intended to show statistical significance of diet and age on starvation resistance in a controlled environment. Given this, here are a few observations/questions:

a) I do not understand the “letters = mean values” labelling at all in Figures 1 and 2 and the information in the caption does not help me in understanding how to interpret this. In figures like this, I much prefer the asterisk “star” method in designating the statistical significance of the mean values between two criteria.

For example:

0.05 > p-value > 0.01 = *

0.01 > p-value > 0.001 = **

0.001 > p-value = ***

Something like the above will help immensely in interpreting the graph. Also make sure that you add the asterisk(s) above a “connector of sorts between the two comparisons (like here: https://www.datanovia.com/en/blog/how-to-add-p-values-onto-basic-ggplots/)

b) In the results, comparisons were made between mated males on full diet and restricted diet, but the figures only show comparisons of mated and virgin males on a specific diet (and same for females). These comparisons could also be visualized in the figures as part C to make it clearer that you are reporting those comparisons as well.

c) I understand that the results are using a common sentence format to compare the data, but the language is repetitive, and it is hard to follow with all the numerical values and statistics being reported within the sentences. I think that a table that reports the values will allow the results section to flow more monthly and be able to make these comparisons more easily by referring to the table if needed. It could be argued that the table might be redundant with the bar graphs, but I think it would be good to visualize the results this way and have a more concise results section.

d) In the paper, it was mentioned that 3200 flies were used. The table mentioned above should also give a count of how many flies of each category were used for each cohort. Show the raw data upon which the medians, means, standard deviations, and standard errors were derived from. This will also boost confidence in the results being reported here.

d) The figures are pixelated and a little rough in terms of low dpi/resolution. I am not sure if it is because of the peer review copy specifically or not, but I would encourage saving these graphs at a much higher resolution to make them easier on the eyes.

e) The statistical methods are appropriate for the comparisons being made here.

Summary:

These figures require substantial work to be more friendly and understandable to the reader.

4) Are the conclusions consistent with the evidence and arguments presented?

a) The conclusions are consistent with the reported results. However, the discussion section places too much emphasis on work done in other insect species, and not enough on possible explanations for the results that were reported. That is not inherently bad, but the authors need to provide a reason for this approach. I am guessing that the work done in other insect species has not been done in the olive fruit fly. If this is true, this needs to be made very clear.

b) Some of these results are not explicitly discussed further. For instance, I am intrigued by how mated males on a full diet were less resistant to starvation in the age cohort 11-20 but more resistant in the age cohort 21-30 (and implies the opposite on restricted diet). I want to know more about what the authors think is happening in olive fruit flies. The authors did talk about lipid contents declining in aging C. capitata flies but did not suggest doing follow-up studies on lipid profiles of these cohorts. I think it would be good to mention this because the adult diet were either yeast/sugar, or sugar alone, and larvae were reared on olives which obviously contains olive oil and other lipids. Could adults be fed lipids and evaluated for changes in starvation resistance? If starvation resistance is increased across the board with adult lipid supplementation, then that would be a significant part in optimizing rearing strategies of sterile insects.

c) Lines 362-368: At this point in the discussion, the reader is given the information that the current sterile insect technique is focused on rearing high numbers of sterilized males for population control. This was not mentioned in the introduction to my knowledge, so the reader may wonder why the authors spent time studying the effects of diet and age on starvation resistance in females. It is important to state very clearly in the discussion that this work is not only applicable to pest management strategies, but also in highlighting the tradeoffs between studying model organisms in laboratory conditions versus natural environments in which many parameters are fluctuating.

Summary:

The discussion seems too general and not focused enough on interpreting the results.

5) Are the cited references mostly recent publications (within the last 5 years) relevant? Does it include an excessive number of self-citations? Please evaluate the ethics statements and data availability statements to ensure they are adequate.

The references are overall current, and no excessive self-citations were seen. The ethics statement and conflict of interests’ statement are adequate. No data availability statements were made.

I found the paper very easy to read in general and left some suggestions in the peer review for minor/moderate fixes. Thank you for your effort in composing the paper in English, it is greatly appreciated.

Reviewer 2 Report

The aim of the study was to estimate the influence of several factors (age, mating status, and pre-test diet) on starvation resistance of the olive fly, Bactrocera oleae, adults. With this aim the authors conducted a very simple laboratory experiment yielded rather straightforward results. The experiments were well planned and conducted; the conclusions made in the Discussion are supported by the data. The results of the study can be used to improve the methods for the control of this important pest. Thus, I think that, generally speaking, the manuscript can be published, although numerous changes and corrections are required (see below).

The problem is that I am not sure that the presented results are sufficient for an article. Yes, indeed, grace to the long (I would say too long when compared with the Results section) Discussion, the total manuscript is large enough. However, the presented study includes only one (very simple) experiment which yielded more or less expected results. Thus, despite of a large total sample size (about 3200 flies) both the amount and the novelty of the presented data are rather limited. Therefore I would suggest to shorten the Discussion and to fit the size of Communication, although it is not mandatory and the ultimate decision is certainly up to the editors.

Lines 8-14: Summary is too short and not informative. On the other hand, some sentences, e.g. “Only deaths were recorded” are not important could be deleted.

Lines 8 and 15: common name “the olive fly” should be mentioned in Summary and Abstract.

Line 10: Please, describe shortly the composition of the full and restricted diets (as in lines 105-106).

Lines 11-12: Please, describe not only the effect of age, but also those of diet and mating status.

Lines 21-22: Again, please, describe shortly the composition of the full and restricted diets. This information is much more important for the readers than, e.g. total sample size (line 17).

Line 27: It would be reasonable to include both common and Latin name of a studied insect in the list of keywords.

Line 87: In this section, not only Methods but also Materials should be described. Please, give information about the laboratory strain / population used for the experiments: place and time of initial collection, methods and conditions of rearing, etc.

Line 123: I think “were” should be replaced: “Data were presented as mean...”

Line 131: But why these data have not been treated by 4-way ANOVA including also sex as a factor? Please, explain.

Lines 203-206 and 263-266: The 2nd sentence of the legends to both figures (“Letters are assigned...”) should be deleted because this information is given in the last sentence (“Bars with different letter...”)

Line 269: I would not say that Bactrocera oleae is really “related to Drosophila melanogaster”: Indeed, both species are from Diptera: Brachycera. However, although common names of both Drosophilidae and Tephritidae are “fruit flies”, they are two quite different (and not very closely related) families.

Figures and particularly labels are two small and difficult to the readers. I would suggest changing the figures by combining the two graphs not horizontally but vertically (A in the top, B in the bottom). This would allow increasing the graph size within the same page size.

Reviewer 3 Report

The manuscript by Balampekou and colleagues, reports an experimental study aimed to examine the starvation resistance of the fruit fly pest Bactrocera oleae, as a function of the adult diet, the mating status and the age of flies.  This was done independently for female and male flies.  The topic of starvation resistance is relevant from theoretical and applied perspectives.  Especially for the application of the sterile insect technique.  It seems to me that the results from this research could potentially make a valuable contribution to the literature on starvation resistance of tephritid fruit flies.  But first, the authors should overcome several weaknesses of their manuscript including basic errors such as repeatedly incorrect writing of scientific names, lack of appropriate citations, a methodological section lacking many important details, and claims with apparently no support from the data.  Overall, the manuscript should improve its structure and line of reasoning/linking of ideas to allow a better understanding of the study and to appreciate the value of the research.  I recommend the authors ask a native English-speaking colleague to review a new and improved version of the manuscript before submission to be sure that there are no English errors.  Below I provide specific comments on these and other points that I believe could help authors improve their manuscript’s quality.

The simple summary should be presented in plain language.  Please try to make it more understandable for the general reader.  For instance, instead of saying "Bactrocera oleae (Rossi) (Diptera: Tephritidae)" (lines 8-9), say something like "the olive fruit fly, a pest of economic importance that threatens olive industry".  You should explain with simple words what you did, what you found and why it is important, leaving aside technical details and specialized language.

Introduction

I found a severe lack of citations to support the arguments presented (e.g., lines 30-31, 34-35, 38-40, 41-44, 55-57, 66-68, 70-71, ...).  Authors must recognize and cite previous work on which they base their statements.

I found that the structure and line of reasoning of the introduction needs improvement.  For instance, in lines 44-46, what is the relevance of this information in the context of starvation resistance?  In lines 58-59, what is the point of giving details about the shape of wings if you do not get back to this topic later in the manuscript?

L 38, Is it common practice to refer to starvation resistance as SR?  I think many readers would prefer to see starvation resistance spelled out in full across the manuscript.

L 44, Drosophila melanogaster with italics and add descriptor of the species.

L 52, Revise "Ceratitis capitatα", and add the author of the species, order and family.

L 53, Write Bactrocera in full because it is the first time you mention this scientific name in the main body of the manuscript, and add the name of the descriptor of the species.

Repeated basic flaws in the spelling of scientific names, such as not using italics, and not spelling out the complete genus name or species author the first time the species is mentioned, reflect an oversight on the part of the authors that do not speak well of the detail and care with which they prepared their manuscript.  In a field of excellence, repeated flaws of this type in a manuscript submitted for peer review are unacceptable.

In its current form, the introduction tries to cover many topics that are not necessarily the focus of the study (e.g., starvation resistance in relation to genetic and physiological mechanisms, starvation resistance in relation to the SIT, starvation resistance in relation to aging, etc.).  I recommend authors to better place the context of their study in the introduction to flush out the significance of their research.  A basic structure that could work to this end is this: (1) first paragraph, introduce the topic and highlight why it is important; (2) second paragraph, relate to current knowledge, what has been done and what needs to be done; (3) third paragraph, introduce your work clearly stating your objectives, and ideally hypothesis and predictions.  The information is somehow already present in the introduction, but the way it is presented needs improvement.

Materials and Methods

The materials and methods section is rather vague consisting only of two sections "2.1. Experimental design" and "2.2. Statistical analysis".  This section needs to be specific and provide all the necessary details to replicate the study.

In the experimental design section, the authors fail to describe the experimental design and instead, they half-describe the experimental protocol and laboratory conditions in which the experiment was performed, and give some elements about the experimental design.

The Materials and Methods section should include, at least, the following elements:

(1) Origin/source/handling of experimental insects

(2) Experimental design with a description of the following: (i) which were the explanatory and response variables, including their levels and measurement scales; (ii) which were the experimental units, the unit of measurement and the unit of response; (iii) how does the experimental units were distributed in time and space (randomization of units); (iv) how many replicates there were per combination of predictor variables; (v) what was the goal of the experimental design

(3) Experimental procedures/protocol: describe the steps you followed in the experiment and the environmental conditions under which the experiment was performed, it is important that you made clear how you measured response variables for others to replicate exactly what you did

(4) Statistical analyses

L 106-107, What is "Ample water"?  Please explain.

L 110, The acronym GLM usually stands for Generalized Linear Model, to avoid confusion, please do not use GLM to refer to General Linear Model.

L 121-122, Please explain how the residuals were tested for normality and equal variance.

In its current form, the methodological description of the experiment lacks many specific details for others to replicate the study exactly how it was done.  I recommend authors be detailed and specific in the description of their methodology, including all the required elements for readers to understand how, exactly, the experiment was conducted.

Results

L 132, It is confusing that the authors indicate they used one-way anova when they are reporting three-way interactions (i.e., three-way anova).

L 133-134, The "simple-simple main effects analysis approach" should be described in the methodology section and appropriate references cited.

It is required that the results indicate the statistics of the full anova model, of specific model terms (i.e., statistics of main and interaction effects), and finally of specific comparisons.

Discussion

The style of the discussion seemed confusing and hard to follow to me. Specifically, in many paragraphs, authors describe or highlight results from previous studies which are cited in many but not all cases (e.g., 275-276, 284-285, 303-305).  So, in some instances, it is difficult to understand if authors are talking about a previous study that they are not citing or if they are discussing their own results. I strongly recommend that authors make correct use of citations of previous studies throughout their manuscript.  In the discussion, the only text that should appear without citations is the one that describes or discusses the results obtained in the present study, proposals for future research, new hypotheses proposed by the authors, etc. In addition, I suggest to the authors that when they are talking about their results, they indicate it at the beginning of the text.  For example, "The results of the present study indicate that...", "In this study, we show that...", "Here we found that...", etc.

L 269-270, What is the rationale/value of starting the discussion by mentioning the relatedness of the olive fruit fly with Drosophila melanogaster?  There is no discussion about this relatedness, so why is it important to mention it at the beginning of the discussion?

L 271, The fact that reproduction is energetically costly is well known in animals, including insects.  You must recognize previous works on the energetic costs of reproduction in tephritid fruit flies in general and in the olive fruit fly in specific.  I am also struggling to find the specific results that show the energetic cost of reproduction you are discussing here.  How did you measured energetic cost?  It is not stated in the manuscript.  Authors should be careful not to mislead readers about findings they did not report in their work.

L 291, Please revise or explain "protein diet".

L 323, Add reference number.

L 361-373, This text is not discussion, it is background information about the SIT.  Should reduce to only a few lines or find a place to include this information in the introduction.

I recommend authors to make their manuscript reviewed by a native English-speaking colleague before submitting to a journal.